# Improved Coresets and Sublinear Algorithms for Power Means in Euclidean Spaces

**Vincent Cohen-Addad**[*]
Google Research, Zurich.

**David Saulpic**[*]
Sorbonne Université, Paris

**Chris Schwiegelshohn**[*]
Aarhus University

## Abstract

In this paper, we consider the problem of finding high dimensional power means: given a set $A$ of $n$ points in $\mathbb{R}^d$, find the point $m$ that minimizes the sum of Euclidean distance, raised to the power $z$, over all input points. Special cases of problem include the well-known Fermat-Weber problem – or geometric median problem – where $z = 1$, the mean or centroid where $z = 2$, and the Minimum Enclosing Ball problem, where $z = \infty$.

We consider these problem in the big data regime. Here, we are interested in sampling as few points as possible such that we can accurately estimate $m$. More specifically, we consider sublinear algorithms as well as coresets for these problems. Sublinear algorithms have a random query access to the set $A$ and the goal is to minimize the number of queries. Here, we show that $\widetilde{O}\left(\varepsilon^{-z-3}\right)$ samples are sufficient to achieve a $(1+\varepsilon)$-approximation, generalizing the results from Cohen, Lee, Miller, Pachocki, and Sidford [STOC '16] and Inaba, Katoh, and Imai [SoCG '94] to arbitrary $z$. Moreover, we show that this bound is nearly optimal, as any algorithm requires at least $\Omega\left(\varepsilon^{-z+1}\right)$ queries to achieve said approximation.

The second contribution are coresets for these problems, where we aim to find find a small, weighted subset of the points which approximates cost of every candidate point $c \in \mathbb{R}^d$ up to a $(1 \pm \varepsilon)$ factor. Here, we show that $\widetilde{O}\left(\varepsilon^{-2}\right)$ points are sufficient, improving on the $\widetilde{O}\left(d\varepsilon^{-2}\right)$ bound by Feldman and Langberg [STOC '11] and the $\widetilde{O}\left(\varepsilon^{-4}\right)$ bound by Braverman, Jiang, Krauthgamer, and Wu [SODA 21].

## 1 Introduction

Large data sets have shifted the focus of algorithm design. In the past, an algorithm might have been deemed feasible if its running time was polynomial in the input size and so a textbook *fast* algorithm can have time complexity for example quadratic. For truly gargantuan data sets, even linear time or nearly linear time algorithms could be considered too slow or requiring too much memory. This led to the emergence of the field of sublinear algorithms: How well can we solve a problem without reading the entire input?

Except for trivial problems, deterministic time sublinear algorithms do not exist. Our primary tool in designing sublinear algorithms is thus the following basic approach:

- Take a uniform sample of the input.
- Run an algorithm on the sample.

Hence, the performance of a sublinear algorithm is often measured in terms of its *query complexity*, i.e. the number of samples required such that we can extract a high quality solution in the second

---

[*]Equal contribution.

35th Conference on Neural Information Processing Systems (NeurIPS 2021).

step above that generalizes to the entire input. Sublinear algorithms have close ties to questions in learning theory and estimation theory, where we are similarly interested in a quality to sample size tradeoff.

A perhaps very fundamental problem of primary importance in machine learning and data analysis is to efficiently estimate the parameters of a distribution. For example, given a distribution $\mathcal{D}$, how many samples do we need to estimate the mean? Even such a simple and basic question has surprisingly involved answers and are still subject to ongoing research (Lugosi & Mendelson (2019); Lee & Valiant (2021)).

In this paper, we investigate the possibility of estimating power means in high dimensional Euclidean spaces. Specifically, given an arbitrary set of points $A$, we wish to determine the number of uniform queries $S$ such that we can extract a power mean $m$ with

$$\text{cost}(m) := \sum_{p \in A} \|p - m\|^z \leq (1 + \varepsilon) \cdot \min_{\mu} \sum_{p \in A} \|p - \mu\|^z,$$

where $\|p\|$ denotes the Euclidean norm of a vector $p$.

The power mean problem captures a number of important problems in computational geometry and multivariate statistics. For example, for $z = 1$, this corresponds to the Fermat-Weber problem also known as the geometric median. For $z = 2$, the problem is to determine the mean or centroid of the data set. Letting $z \to \infty$, we have the *Minimum Enclosing Ball* (MEB), where one needs to find the Euclidean sphere of smallest radius containing all input points.

For $z > 2$, the problem is not as well studied, but it still has many applications. First, higher powers allows us to interpolate between $z = 2$ and $z \to \infty$, which is interesting as the latter admits no sublinear algorithms[2]. Skewness (a measure of the asymmetry of the probability distribution of a real-valued random variable about its mean) and kurtosis (a measure of the "tailedness" of the probability distribution) are the centralized moments with respect to the three and the four norms and are frequently used in statistics. The power mean is a way of estimating these values for multivariate distributions.

Another application is when dealing with non-Euclidean distances, such as the Hamming metric, coreset constructions for powers of $z$ can be reduced to coreset constructions for powers $2z$. So for example if we want the mean in Hamming space, we can reduce it to the $z = 4$ case /in squared Euclidean spaces Huang & Vishnoi (2020).

These problems are convex and thus can be approximated in the near-linear time efficiently via convex optimization techniques. However, aside from the mean ($z = 2$), doing so in a sublinear setting is challenging and to the best of our knowledge, only the mean and the geometric median ($z = 1$) are currently known to admit nearly linear time algorithms.

Our main result is:

**Theorem 1.** *There exists an algorithm that, with query complexity $O(\varepsilon^{-z-3} \cdot polylog(\varepsilon^{-1}) \log^2 1/\delta)$, computes a $(1 + \varepsilon)$ approximate solution to the high dimensional power mean problem with probability at least $1 - \delta$.*

A key component in designing this algorithm is a novel analysis for *coresets* for these problems. Coresets are succinct summaries that approximate cost for any center solution $c$.

**Theorem 2.** *For any set of points in $d$-dimensional Euclidean space, there exists an $\varepsilon$-coreset for the high dimensional power mean problem of size $\tilde{O}(\varepsilon^{-2} \cdot 2^{O(z)})$.*

With the exception of the mean, previous coresets for these problems achieved bounds $O(\varepsilon^{-2} \cdot 2^{O(z)} \cdot \min(d, \varepsilon^{-2}))$ (Cohen-Addad et al. (2021); Braverman et al. (2021); Feldman & Langberg (2011)), or had weaker guarantees such as merely approximating an optimal solution or requiring removal of outliers from the data set.

Comparing the bounds in Theorem 1 and Theorem 2, one may question whether the exponential dependency in $z$ is necessary for computing an approximation. Indeed, previous sublinear algorithms

---

[2]To see this, we place $n - 1$ points at 0 and one point with probability $1/2$ at 1 and with probability $1/2$ at 0. Any $2 - \varepsilon$ approximation can distinguish between the two cases, but this clearly requires us to query $\Omega(n)$ points.

for both the geometric median and the mean had a query complexity of $O(\varepsilon^{-2})$, and thus matched our coresets bounds. Unfortunately, we show that the exponential dependency in the power is indeed necessary even in a single dimension:

**Theorem 3.** *For any $\varepsilon > 0$ and $z$, any algorithm that computes with probability more than $4/5$ a $(1 + \varepsilon)$-approximation for a one-dimensional power mean has query complexity $\Omega(\varepsilon^{-z+1})$.*

Hence, up to constants in the exponent, our sublinear algorithm is tight. Moreover, the algorithm is very simple to implement and performs well empircally.

## 1.1 Techniques

While stochastic gradient descent has been used for a variety of center-based problems (Clarkson et al. (2012); Cohen et al. (2016)), it is difficult to apply it for higher powers. Indeed, Cohen et al. (2016) remark in their paper that even for the mean[3] ($z = 2$) their analysis does not work as the objective function is neither Lipschitz, nor strictly convex.

Hence, one needs to use new tools. A natural starting point is to use techniques from coresets, as they allow us to preserve most of the relevant information, using a substantially smaller number of points. Unfortunately, coresets have a drawback: the sampling distributions used to construct coresets is non-uniform and therefore difficult to use in a sublinear setting. The first step is therefore to design coresets from uniform sampling. To do this, we use and improve upon a technique originally introduced by Chen (2009). Chen showed that, given a sufficiently good initial solution $q$, one can partition the points into rings exponentially increasing radii such that the points cost the same, up to constant factors. Thereafter, taking a uniform sample of size $\tilde{O}(d \cdot \varepsilon^{-2})$ from each ring produces a coreset. Since there are at most $O(\log n)$ rings in the worst case, this yields a coreset of size $\tilde{O}(d \cdot \varepsilon^{-2} \cdot \log n)$.

To realize these ideas in a sublinear setting, we are now faced with a number of challenges. First, rings that are particularly far from $q$ may contain very few points. This makes is difficult for a sublinear algorithm to access them. Second, partitioning the points into rings depends on the cost of $q$. It is very simple to construct examples where estimating the cost of an optimal power center requires $\Omega(n)$ many queries. Finally, this analysis loses both factors $\log n$ and $d$, which we aim to avoid.

We improve and extend this framework in two ways. First, we show that it is sufficient to only consider $O(\log \varepsilon^{-1})$ many rings, which, in of itself, already removes the dependency on $\log n$. Moreover, we show that it is possible to simply ignore any ring containing too few points, i.e. any ring with less than $\varepsilon^{z+O(1)} \cdot n$ points may be discarded. The intuition is that, while rings with few point may contribute significantly to the cost, these points do not influence the position of the optimal center by much. Thus, using a number of carefully chosen pruning steps, we show how to reduce both the problem of obtaining a sublinear algorithm as well as obtaining coresets to sampling from a select few rings containing many points.

The second improvement directly considers an improved analysis of sampling from rings. The standard way to prove a coreset guarantee is to show that using $s \log 1/\delta$ samples we preserve the cost of a single solution with probability $1 - delta$. If there exist $T$ solutions then we set $\delta = 1/T$ and have obtained a coreset, which we call the "naive" union bound. This works in certain cases such as finite metrics, but is insufficient if we have infinitely many solutions such as Euclidean spaces. The simplest way to improve over the naive union bound is to discretize the space and then apply the naive union bound on the discretization. In literature this is sometimes called an $\varepsilon$-net bound. This can be optimal or close to optimal for certain metrics, but so far these arguments have only lead to $\tilde{O}(\varepsilon^{-2} \min(d, \varepsilon^{-2}))$ bounds for coresets in Euclidean spaces.

Instead of applying a union bound over the discretization in "one shot", we apply a union bounded over a nested sequence of increasingly better discretizations. Essentially, instead of only using a set of centers $C_\varepsilon$ as a substitute for all centers is $\mathbb{R}^d$, we use centers $c^h \in C_h$ for different values of $h \in \{1, \ldots, \log 1/\varepsilon\}$. In literature, this is known as chaining, see Nelson (2016) for a survey. We can then write, for any input point set $P$, $\sum_{p \in P} \text{cost}(p, c) = \sum_{p \in P} \sum_{h \geq 0} \text{cost}(p, c^{h+1}) - \text{cost}(p, c^h)$, where $\text{cost}(p, c^0)$ is defined to be 0 and $|\text{cost}(p, c^h) - \text{cost}(p, c)| \leq 2^{-h} \text{cost}(p, c)$. We now only

---

[3]For the special case of the mean, Inaba et al. Inaba et al. (1994) observed that $O(\varepsilon^{-2})$ samples are nevertheless sufficient.

apply the naive union bound for successive summands, i.e. we approximate $\sum_{p \in P} \text{cost}(p, c^{h+1}) - \text{cost}(p, c^h)$ up to an error of $\varepsilon \cdot \sum_{p \in P} \text{cost}(p, c)$.

Essentially, the idea is to use a sequence of solutions $c_h$ that approximate a candidate solution $c$ with increasing accuracy as $h \to \infty$. Approximating the cost of a candidate center $c$ can then be written as a telescoping sum of solutions $c_h$, i.e. $\text{cost}(c) = \sum_{h=0}^{\infty} \text{cost}(c_{h+1}) - \text{cost}(c_h)$, where $c_h$ is a solution that has *distance* to $c$ of the order $2^{-h}$ and $\text{cost}(c_0) := 0$. The notion of distance between candidate solutions is perhaps most easily understood by imagining a solution to be a cost vector $v^c$ where the $i$th entry corresponds to the cost of the $i$th point of $A$ when using $c$ as a candidate center, i.e., $v_i^c = \text{cost}(p_i, c)$. Informally, we consider two solutions $c$ and $c'$ to have distance at most $2^{-h}$ if $|\|p_i - c\|^z - \|p_i - c'\|^z| \leq 2^{-h} \cdot (\|p_i - c\|^z + \|p_i - c'\|^z)$ for all points $p_i \in A$.

## 1.2 Related Work

**Sublinear Approximation for Clustering:** A number of sublinear algorithm are known for clustering problems. For $k$-Median, under the constraint that the input space has a small diameter, a constant factor approximation is known Czumaj & Sohler (2007). Ben-David (2007) proposed a different set of conditions under which a sublinear algorithm for $k$-median and $k$-means exists. Other approximations, with different constraint are also known: for instance, Meyerson et al. (2004) give an algorithm achieving a $O(1)$-approximation in time $\text{poly}(k/\varepsilon)$ for discrete metrics, when each cluster has size $\Omega(n\varepsilon/k)$. For the 1-median problem, this assumption is always satisfied, and their algorithm gives a constant factor approximation in constant running time. The algorithm by Cohen et al. (2016) produces a $(1 + \varepsilon)$-approximation in time $\min(nd \log^3(n/\varepsilon), d/\varepsilon^2)$ for Euclidean spaces of dimension $d$. Ding (2020, 2021) and Clarkson et al. (2012) showed how to obtain sublinear algorithms for the minimum enclosing ball problem assuming that either the algorithm is allowed to drop a fraction of the points, or with an additive error. For the unconstrained version of the problem, no sublinear time algorithm is possible. For the $k$-means problem, Bachem et al. (2016) showed how to approximate the $k$-means++ algorithm in sublinear time. To our knowledge, no algorithm is known for higher distance powers $z$.

**Coreset Constructions:** A coreset is a weighted subset of the input points, such that the cost of any clustering is the same on the coreset than on the input points, up to a $(1 \pm \varepsilon)$ factor. The arguably most widely studied problem is coresets for the $k$ clustering problems with powers of distances. Following a long line of work Bachem et al. (2018); Becchetti et al. (2019); Braverman et al. (2021); Chen (2009); Cohen-Addad et al. (2021); Feldman & Langberg (2011); Feldman et al. (2020); Feng et al. (2021); Fichtenberger et al. (2013); Har-Peled & Kushal (2007); Har-Peled & Mazumdar (2004); Huang & Vishnoi (2020); Langberg & Schulman (2010); Sohler & Woodruff (2018), we now know that coreset of size $\tilde{O}(k\varepsilon^{-4} \cdot 2^{O(z)} \cdot \min(k, \varepsilon^{-\max z-2, 0}))$ exist. In a low-dimensional regime, this may be improved to $\tilde{O}(k^2 d \varepsilon^{-2} 2^{O(z)})$ and $\tilde{O}(kd\varepsilon^{-2z})$ Feldman & Langberg (2011). For the special case of clustering with a single center, this yields the state of the art $\tilde{O}(\varepsilon^{-4})$ bound due to Braverman et al. (2021). For the Minimum Enclosing Ball problem and its generalizations, these algorithms yield no space saving, although sketches of weaker gaurantees of size $O(\varepsilon^{-1})$ exist (Badoiu & Clarkson (2008)). Given a coreset, it is easy to compute a $(1 + \varepsilon)$-approximation in time independent of $n$: one just needs to find a $(1 + \varepsilon)$-approximation on the coreset points.

Coresets have also been studied for many other problems: we cite non-comprehensively fair clustering Cohen-Addad & Li (2019); Huang et al. (2019); Schmidt et al. (2019) determinant maximization Indyk et al. (2020), diversity maximization Ceccarello et al. (2018); Indyk et al. (2014) logistic regression Huggins et al. (2016); Munteanu et al. (2018), dependency networks Molina et al. (2018), or low-rank approximation Maalouf et al. (2019). The interested reader is referred to the recent surveys for more information Feldman (2020); Munteanu & Schwiegelshohn (2018).

## 1.3 Organization

We prove our key sampling result in Section 2. We follow up on that result by applying it to the sublinear setting (Section 3) and to coresets (Section 4). We conclude with a short experimental evaluation. Due to space constraints, all proofs are included in the supplementary material.

## 1.4 Preliminaries

We denote the Euclidean distance of a vector $x$ by $\|x\| := \sqrt{\sum_{i=1}^{d} x_i^2}$. Similarly, we define the Hamming norm $\|x\|_1 = \sum_{i=1}^{d} |x_i|$. For a set of points $A$, we say that $\|A\|_0$ is the distinct number

of points. Note that if $A$ contains multiplicities $|A| \neq \|A\|_0$. We write $\tilde{O}(x)$ to denote $O(x \cdot \log^a x)$, where $a$ is any constant. For any set $A$ and candidate solution $c$, we defined $\text{cost}(A, c) = \sum_{p \in A} \|p - c\|^z$. If the set is clear from context, we simply write $\text{cost}(c)$. We frequently use the following generalized triangle inequality, see Cohen-Addad & Schwiegelshohn (2017); Makarychev et al. (2019) for proofs and similar statements.

**Lemma 1** (Triangle Inequality for Powers). *Let $a, b, c$ be an arbitrary set of points in a metric space with distance function $d$ and let $z$ be a positive integer. Then for any $\varepsilon > 0$*

$$d(a, b)^z \leq (1 + \varepsilon)^{z-1} d(a, c)^z + \left(\frac{1+\varepsilon}{\varepsilon}\right)^{z-1} d(b, c)^z$$

$$|d(a, b)^z - d(a, c)^z| \leq \varepsilon \cdot d(a, c)^z + \left(\frac{z+\varepsilon}{\varepsilon}\right)^{z-1} d(b, c)^z.$$

We also use the fact that uniform sampling is efficient to approximate the density: we review details on sampling in bounded VC dimension in the supplementary material.

**Lemma 2** (Li et al. (2001)). *Given a range space $(X, \mathcal{R})$ with VC-dimension $d$, an constants $\varepsilon, \delta, \eta$, and a uniform sample $S \subset X$ of size at least $O(\frac{1}{\eta \cdot \varepsilon^{-2}}(d \log 1/\eta + \log 1/\delta))$, we have for all ranges $R \in \mathcal{R}$ with $|X \cap R| \geq \eta \cdot |X|$*

$$\left| \frac{|X \cap R|}{|X|} - \frac{|S \cap R|}{|S|} \right| \leq \varepsilon \cdot \frac{|X \cap R|}{|X|}$$

*and for all ranges $R \in \mathcal{R}$ with $|X \cap R| \leq \eta \cdot |X|$*

$$\left| \frac{|X \cap R|}{|X|} - \frac{|S \cap R|}{|S|} \right| \leq \varepsilon \cdot \eta$$

*with probability at least $1 - \delta$.*

The only range space we will consider is the one induced by Euclidean spheres centered around a single fixed point. The VC dimension induced of this range space is 2, which seems to be a well known fact, although we could not find a reference. For completeness, we added a short proof in the supplementary material.

First, we recall the commonly used coreset definition for clustering problems in Euclidean spaces.

**Definition 1.** *Let $A$ be a set of points in $\mathbb{R}^d$. Then a set $\Omega$ is a strong $(\varepsilon, z)$-coreset if there exists a weight function $w : \Omega \to \mathbb{R}^+$ and a constant $\kappa$ such that for any point $c$*

$$\left| \sum_{p \in A} \|p - c\|^z - \left( \sum_{p \in S} w_p \|p - c\|^z + \kappa \right) \right| \leq \varepsilon \cdot \sum_{p \in A'} \|p - c\|^z.$$

*We say that $\Omega$ is a weak $(\varepsilon, z)$-coreset if for some $\alpha \in [0, 1]$ any point satisfying $\sum_{p \in \Omega} w_p \cdot \|p - c'\|^z \leq (1 + \alpha \cdot \varepsilon) \text{argmin}_{c \in \mathbb{R}^d} \sum_{p \in \Omega} w_p \cdot \|p - c\|^z$ also satisfies $\sum_{p \in A} \|p - c'\|^z \leq (1 + \varepsilon) \text{argmin}_{c \in \mathbb{R}^d} \sum_{p \in A} \|p - c\|^z$.*

The difference between the two notions is that strong coresets give a guarantee for all candidate centers, whereas the weak coreset guarantee only applies for the optimum. In an offline setting, there is no difference in the size for our construction for either guarantee. In the sublinear setting, we will be satisfied with a weak coreset as it can be obtained with a nearly optimal query complexity.

## 2 Uniform Sampling Routine

In this section, we outline the proof of our basic sampling subroutine. We assume that we are given a point $q$, and a set of points $R$ such that for all $p, p' \in R$

$$\|p - q\|^z \leq 2 \cdot \|p' - q\|^z.$$

In the following sections, we refer to $R$ as a ring. Under this assumption, the following claim holds.

**Theorem 4.** *Let $q$ and $R$ be defined as above and let $\Omega$ be a uniform random sample consisting of $s \in \tilde{O}(\varepsilon^{-2} \cdot \log 1/\delta)$ points. Then with probability at least $1 - \delta$, we have for all candidate centers $c$*

$$\left| cost(R,c) - \sum_{p \in \Omega} \frac{|R|}{s} \cdot \|p - c\|^z \right| \leq \varepsilon \cdot (cost(R,q) + cost(R,c)).$$

We prove this theorem, first by proving the following slightly simpler result and then showing how to apply the result recursively to remove dependency in $\|R\|_0$:

**Lemma 3.** *Let $q$ and $A$ be defined as above and let $\Omega$ be a uniform random sample consisting of $s \in \tilde{O}(\varepsilon^{-2} \cdot \log \|R\|_0 \cdot \log 1/\delta)$ points. Then with probability at least $1 - \delta$, we have for all candidate centers $c$ that $\left| cost(R,c) - \sum_{p \in \Omega} \frac{|R|}{s} \cdot \|p - c\|^z \right| \leq \varepsilon \cdot (cost(R,q) + cost(R,c))$.*

To set up a chaining analysis, we require two ingredients: (1) a notion of nets and (2) a Gaussian process. We focus on the latter first. For any point $c$, let $v^c$ be the $|R|$-dimensional vector, henceforth called a *cost vector*, such that $v_{p_i} = \|p_i - c\|^z$ for some arbitrary but fixed ordering of the points in $R$. Note that $\|v\|_1 = cost(R,c)$. Let $p_j \in \Omega$ with $j \in \{1, \ldots, s\}$ be the $j$th point of the sample. Observe for any cost vector $v$ induced by some center $c$, we have

$$\mathbb{E}_\Omega \left[ \frac{n}{s} v_{p_j} \right] = \sum_{p \in R} \|p - c\|^z = cost(R,c).$$

We now symmetrize the expectation. Let $g_1, \ldots g_s$ be standard Gaussian random variables, i.e. Gaussians with mean 0 and variance 1. Then we have for any collection of cost vectors $\mathbb{N}$ (see Appendix B.3 of Rudra & Wootters (2014))

$$\mathbb{E}_\Omega \sup_{v \in \mathbb{N}} \left| \frac{\sum_{j=1}^s \frac{|R|}{s} v_{p_j} - \|v\|_1}{cost(R,q) + \|v\|_1} \right| \leq \sqrt{2\pi} \mathbb{E}_\Omega \mathbb{E}_g \sup_{v \in \mathbb{N}} \left| \frac{\sum_{j=1}^s \frac{|R|}{s} v_{p_j} \cdot g_j}{cost(R,q) + \|v\|_1} \right| \tag{1}$$

Note that the first term in Equation 1 is the expected maximum deviation from the (normalized) expected cost of the sample, if the cost vectors are induced by centers. In other words, if $\sup_{c \in \mathbb{R}^d} \left| \frac{\sum_{j=1}^s \frac{|R|}{s} \|p_j - c\|^z - cost(R,c)}{cost(R,q) + cost(R,c)} \right| \leq \varepsilon$, we have the desired coreset guarantee. Our cost vectors will not be induced by centers for technical reasons, but are in a well-defined sense close enough such that it will be enough to bound the deviation for these approximate cost vectors to obtain a bound for all center induced cost vectors. Introducing Gaussians is standard in this type of analysis as concentration bounds typically used for weighted Bernoulli random variables such as Hoeffding or Bernstein are too weak. Our goal is therefore to show that $\mathbb{E}_\Omega \mathbb{E}_g \sup_{c \in \mathbb{R}^d} \left| \frac{\sum_{j=1}^s \frac{|R|}{s} \|p_j - c\|^z \cdot g_j}{cost(R,q) + cost(R,c)} \right| \leq \frac{\varepsilon}{\sqrt{2\pi}}$.

We now define the nets we will use.

**Definition 2.** *Let $R$ be a set of points and let $q$ be a candidate solution. For $\beta > 0$, we say that a set of vectors $\mathbb{N}_\beta$ is a $\beta$-net of $R$, if there exists a vector $v' \in \mathbb{N}_\beta$ such that for every point $p \in R$ with $\|p - c\| \leq \frac{8z}{\varepsilon} \cdot \|p - q\|$, we have*

$$\left| \|p - c\|^z - v_p' \right| \leq \beta \cdot \left( \|p - c\|^z + \|p - q\|^z \right).$$

We need to show the following three properties:

**1.** By bounding $\mathbb{E}_\Omega \sup_{v \in \mathbb{N}_{\varepsilon/10}} \left| \frac{\sum_{j=1}^s \frac{|R|}{s} v_{p_j} - \sum_{p \in R_i} v_p}{cost(R,q) + cost(R,c)} \right|$, we can also bound

$\mathbb{E}_\Omega \sup_{c \in \mathbb{R}^d} \left| \frac{\sum_{j=1}^s \frac{|R|}{s} \|p_j - c\|^z - cost(R,c)}{cost(R,q) + cost(R,c)} \right|$

**2.** There exist $\beta$-nets of size $\exp \left( z^2 \log \|R\|_0 \cdot \beta^{-2} \cdot \log \frac{1}{\varepsilon \cdot \beta} \right)$.

**3.** Let $v$ be a cost vector written as a telescoping sum $v = \sum_{i=1} \infty v_i - v_{i-1}$ of cost vectors from $2^i$-nets. Let $v', v''$ be cost vectors from successive $\beta$ and $\beta/2$ nets $\mathbb{N}_\beta$ and $\mathbb{N}_{\beta/2}$, respectively. By bounding the term $\mathbb{E}_\Omega \mathbb{E}_g \sup_{v'-v''} \left| \sum_{j=1}^s \frac{|R|}{s} |v'_{p_j} - v''_{p_j}| \cdot g_j \right|$, we can also bound

$$\mathbb{E}_\Omega \mathbb{E}_g \sup_{v \in \mathbb{N}_\varepsilon} \left| \sum_{j=1}^s \frac{|R|}{s} v_{p_j} \cdot g_j \right|$$

We start with item 1 via the following lemma.

**Lemma 4.** *Let $\mathbb{N}_{\varepsilon/10}$ be an $\varepsilon/10$-net of $R$. Then if $\sup_{v \in \mathbb{N}_{\varepsilon/10}} \frac{\left| \sum_{j=1}^s \frac{|R|}{s} v_{p_j} - \|v\|_1 \right|}{cost(R,q)+cost(R,c)} \leq \frac{\varepsilon}{10}$ we have*
$\sup_{c \in \mathbb{R}^d} \frac{\left| \sum_{j=1}^s \frac{|R|}{s} \|p_j - c\|^z - cost(R,c) \right|}{cost(R,q)+cost(R,c)} \leq \varepsilon$ *for all $c \in \mathbb{R}^d$.*

To prove item 2, we use terminal embeddings Elkin et al. (2017); Mahabadi et al. (2018); Narayanan & Nelson (2019), defined as follows.

**Definition 3** (Terminal Embeddings). *Let $A$ be a set of points in $\mathbb{R}^d$. A mapping $f : \mathbb{R}^d \to R^k$ is a terminal embedding if for all $p \in A$ and all $c \in \mathbb{R}^d$*

$$(1 - \varepsilon) \cdot \|p - c\|_2 \leq \|f(p) - f(c)\|_2 \leq (1 + \varepsilon) \cdot \|p - c\|_2.$$

The guarantee of a terminal embedding is very similar to the guarantee of the famous Johnson Lindenstrauss lemma, but stronger in one crucial detail. A terminal embedding preserves not only the distances between the points of $A$ but also the distance between an arbitrary point in $\mathbb{R}^d$ and any point of $A$. Despite this stronger guarantee, the target dimension of terminal embedding is in fact no worse than that of the Johnson Lindenstrauss lemma. Specifically:

**Theorem 5** (Narayanan & Nelson (2019)). *For any $n$ point-set $A \subset \mathbb{R}^d$, there exists a terminal embedding $f : \mathbb{R}^d \to \mathbb{R}^k$ with $k \in \gamma \cdot \varepsilon^{-2} \log n$ for some absolute constant $\gamma$.*

We now use the terminal embeddings to show that small nets exist.

**Lemma 5.** *Let $R$ and $q$ be defined as above. Then for every $\beta > 0$, there exists a $\beta$-net of $R$ of size at most $\exp(\gamma \cdot z^3 \beta^2 \log \|R\|_0 \cdot \log \varepsilon^{-1})$, where $\gamma$ is an absolute constant.*

We now move onto item 3.

**Lemma 6.** *Let $R$ and $q$ be defined as above and let $\Omega$ be a uniform sample consisting of $s$ points. Then for any point $c$ and $s \geq \eta \cdot z^3 2^{8z} \cdot \varepsilon^{-2} \cdot \log \|R\|_0 \cdot \log^3 \varepsilon^{-1}$ for some absolute constant $\eta$, we have*
$\mathbb{E}_\Omega \mathbb{E}_g \sup_{c \in \mathbb{R}^d} \left| \frac{\sum_{j=1}^s \frac{|R|}{s} \|p_j - c\|^z \cdot g_j}{cost(R,q)+cost(R,c)} \right| \leq \varepsilon$. *Moreover, if $s \geq \eta \cdot z^3 2^{8z} \cdot \varepsilon^{-2} \cdot \log \|R\|_0 \cdot \log^4 \varepsilon^{-1} \log 1/\delta$*
*for some absolute constant $\eta$, then we have $\sup_{c \in \mathbb{R}^d} \left| \frac{\sum_{j=1}^s \frac{|R|}{s} \|p_j - c\|^z \cdot g_j}{cost(R,q)+cost(R,c)} \right| \leq \varepsilon$, with probability at least*
$1 - \delta$.

The proof of Lemma 3 is now a direct consquence of Lemma 6 and Equation 1. To finally prove Theorem 4, we apply Lemma 6 recursively. The result is essentially a special case of Theorem 3.1 from Braverman et al. (2021).

**Lemma 7.** *Let $R$ and $q$ be defined as above. Suppose a uniform sample of size $s \in \tilde{O}(\Gamma \cdot \log \|R\|_0 \cdot \log 1/\delta)$ satisfies with probability at least $1 - \delta$ for all candidate centers $c$*

$$\left| cost(R,c) - \sum_{p \in \Omega} \frac{|R|}{s} \cdot \|p - c\|^z \right| \leq \varepsilon \cdot (cost(R,q) + cost(R,c)).$$

*Then a uniform sample of size $\tilde{O}(\Gamma \cdot \log 1/\delta)$ achieves the same guarantee.*

## 3 Sublinear Algorithm and Analysis

We first describe an algorithm that succeeds with constant probability. This probability can be amplified (non-trivially) using independent repetition. In the following we will use parameters $\beta$ and $\eta$ that depends on $\varepsilon$ which we will specify later. We let $A$ be the set of input points.

---
**Algorithm 1** Sublinear Algorithm for Power Means
---
1. Sample a random point $q$.
2. Sample a set $S$ of $O(\varepsilon^{-z-3} \cdot \text{poly}(\varepsilon^{-1}))$ points uniformly at random.
3. Compute the maximum distance $d$ such that there exist $2/3 \cdot \varepsilon \cdot \eta \cdot |S|$ points with distance at least $d$ from $q$. Discard all points at distance greater than $d$.
4. Define rings $R_i$ such that $R_i \cap S$ contains all the points at distance $(d \cdot 2^{-i}, d \cdot 2^{-i+1}]$ from $q$, with $i = \{1, \dots, \beta\}$.
5. If $|R_i \cap S| < \varepsilon\eta \cdot |S|$, remove all points in $R_i \cap S$ from $S$.
6. Define $\hat{R}_i = n \cdot \frac{|R_i \cap S|}{|S|}$. Weigh the points $R_i \cap S$ by $\frac{\hat{R}_i}{|R_i \cap S|}$.
7. Solve the problem on the (weighted) set $S$.
---

Our goal is to show that $S$ satisfies a weak coreset guarantee. Specifically, we will show that $S$ has the property that for all points $A' \subseteq A$ at distance at most $d$ (defined in Algorithm 1) from $q$ there exists weight function $w : S \to \mathbb{R}^+$ such that $S$ is a strong coreset for $A'$. We then show that a strong coreset for $A'$ is also weak coreset for $A$, i.e. we preserve the cost of all points in $A'$, and we preserve the optimum for $A$.

In order to call Theorem 4 on only few rings, we now show that the loss incurred by only considering $A'$ is indeed negligible.

**Pruning Lemmas** We first show that we can safely discard points that are sufficiently far away, as parameterized by $\gamma$. We recall the meaning of parameters: $\alpha$ is the approximation-factor of the initial solution, $\beta$ is such that points at distance closer than $2^\beta d$ can be merged to the center (see Lemma 9), $\eta$ is such that rings with less than an $\varepsilon\eta$-fraction of the points can be discarded (see Lemma 10)

**Lemma 8.** *Suppose we are given an $\alpha$-approximate center $q$. Let $B(q, r)$ be the ball centered at $q$ with radius $r = 4 \cdot \left(\frac{2\alpha \cdot OPT}{n}\right)^{1/z}$. Then the following two statements hold.*

1. *Any $\alpha$-approximate center is in $B(q, r)$.*

2. *For any two points $c, c' \in B(q, r)$ and for any point $p$ with $\|p - q\|^z > \gamma \cdot r^z$ with $\gamma > \varepsilon^{-z} \cdot (12z)^z$, we have $\|p - c\|^z \leq (1 + \varepsilon) \cdot \|p - c'\|^z$.*

Unfortunately, we are not given a knowledge of $OPT$ a priori. We therefore have to describe how to implement this lemma in a sublinear fashion. For this, we observe the following. First, the point $q$ is an $2^z\alpha$ approximation with probability at least $1 - 1/\alpha$. Second, the number of points that cost more than $\gamma \cdot 2^z\alpha \cdot \frac{OPT}{n}$ is at most $1/\gamma \cdot n$. Combining this observation with Lemma 2 ensures that we are not considering any points that are too far away. Since it is difficult to determine $\frac{OPT}{n}$ in a sublinear fashion, we will rely on additional pruning arguments for points that are close to $q$.

We proceed here in two steps. First, we consider the points that are very close to $q$. Second, we will show that the rings which contain too few points to be efficiently sampled have an overall negligible contribution to the cost.

**Lemma 9.** *Suppose that $q$ is an $\alpha$-approximate solution. Let $A_{near} \subset A$ be a set of points with cost at most $(\varepsilon/(\alpha 5z))^z \cdot \frac{OPT}{n}$. Let $\hat{A} = (1 \pm \varepsilon)|A_{near}|$ Then for any candidate solution $c$ we have*

$$\left| \hat{A} \cdot \|q - c\|^z - \sum_{p \in A_{near}} \|p - c\|^z \right| \leq \varepsilon/\alpha \cdot \left( \sum_{p \in A_{near}} \|p - c\|^z + OPT \right).$$

For rings with few points, we have the following.

**Lemma 10.** *Suppose that $q$ that is an $\alpha$-approximate solution. Let $R_{cheap}$ the union of rings with $|R_i \cap A| < \varepsilon \cdot \eta \cdot n$ and with radius at most $4(\gamma \cdot \frac{\alpha \cdot OPT}{n})^{1/z}$, where $\gamma$ is given by Lemma 8. Then, for any candidate solution $c$*

$$\sum_{p \in R_{cheap} \cap A} \|p - c\|^z \leq \varepsilon \cdot 4^{z+1} \cdot \beta \cdot \eta \cdot \gamma \cdot \alpha \cdot \sum_{p \in A} \|p - c\|^z.$$

While we will defer the exact parameterization to later, observe that for $\beta \in O(\log 1/\varepsilon)$, $\alpha \in 2^{O(z)}$ and $\eta \in O(\varepsilon^z)$ this entire sum can be bounded by $O(\varepsilon) \cdot OPT$.

The next lemma states that we can use these pruning results in a sublinear fashion.

**Lemma 11.** *Let $q$ be a point that is an $\alpha$-approximation and let $S$ be a uniform sample consisting of $O\left(\alpha \cdot \eta^{-1} \cdot \varepsilon^{-3} polylog(\varepsilon^{-1} \cdot \delta^{-1})\right)$ points. Then with probability at least $1 - \delta$ for all rings $R_i$,*

$$|R_i \cap A| - \varepsilon \cdot \max\left(n \cdot \eta, |R_i \cap A|\right) \leq \frac{|R_i \cap S| \cdot n}{|S|} \leq |R_i \cap A| + \varepsilon \cdot \max\left(n \cdot \eta, \ |R_i \cap A|\right).$$

*Furthermore, let $d$ as in the algorithm, i.e., such that $\frac{2}{3} \cdot \varepsilon \cdot \eta \cdot |S| \leq \left|S \setminus (B(q,d) \cap S)\right|$. Then $d < \left(3(\varepsilon\eta)^{-1} \cdot \frac{\alpha \cdot OPT}{n}\right)^{1/z}$*

Since we do not know $r$, we also do not know which of the rings with $i < 0$, if any, satisfy $2^{i+1} \cdot d > (\varepsilon/\alpha 5z) \cdot (\eta/3\alpha)^{1/z}$. However, the maximum number of these rings is at most $\log(\frac{5\alpha z}{\varepsilon} \cdot \left(\frac{3\alpha}{\eta}\right)^{1/z})$, which will turn out to be of the order $O(\log \varepsilon^{-1})$. For all of the rings that are not light, i.e. we cannot discard or snap to $q$, we now use Theorem 4. To ensure that we can call Theorem 4, we invoke Lemma 2 for every ring.

**Probability Amplification**  While the aforementioned algorithm is guaranteed to produce a $(1+\varepsilon)$-approximation with constant probability, amplifying this is non-trivial. Indeed, when running the algorithm multiple times, it is not clear how to distinguish a successful run from an unsuccessful one. The main issue in amplifying the probability lies in the initial solution $q$, as any invocation of Lemma 2 or Theorem 4 allows us to control the failure probability. The simplest way to achieve a success probability $1 - \delta$ is to condition on $\|q - c\|^z \leq \delta \cdot \frac{OPT}{n}$. Unfortunately, this makes $\varepsilon$ dependent on $\delta$, which significantly increases the sampling complexity.

Instead, we use the following algorithm. We sample $m \in O(\log 1/\delta)$ points $q_1, \ldots q_m$ uniformly at random. For each point, we additionally sample $O(\varepsilon^{-2}(\log 1/\varepsilon + \log 1/\delta))$ points $S_{q_i}$. For $q_i$, let $d_i$ denote the minimum radius such that the points in $\frac{n}{|S_{q_i}|} \cdot |B(q_i, d_i) \cap S_{q_i}| > \frac{n}{2}$. We output the point with minimal $d_i$.

The following lemma shows that this point is, with probability at least $1 - \delta$, a $8^z$ approximation.

**Lemma 12.** *Given query access to $A$, we can identify with probability $1 - \delta$ a $8^z$-approximate solution using $O(\varepsilon^{-2}(\log 1/\varepsilon + \log 1/\delta) \log 1/\delta)$ samples.*

To achieve an overall success probability of $1 - \delta$, we only need to sample from non-cheap rings. Thereafter, a high probability bound can be obtained by applying Theorem 4 applied to all $R_i$. The range space induced by rings centered around a single point $q$ has constant VC dimension. Hence, Lemma 2 guarantees that a constant size sample will allow us to distinguish cheap rings from non-cheap ones.

## 4    Improved Coreset Constructions

Our algorithm is as follows. First, we compute a point $q$ that is a reasonably good approximation to the optimum[4]. In the following, let $\alpha = \frac{\text{cost}(q)}{OPT}$. Let $\Delta = \frac{\text{cost}(q)}{n}$ be the average cost of the input points when clustering them to $q$. We now partition the points into rings, defined as follow: $R_i = \{p \in A \mid \left(\frac{\varepsilon}{2z}\right)^z \cdot \frac{1}{\alpha} \cdot 2^i \cdot \Delta \leq \|p - q\|^z \leq \left(\frac{\varepsilon}{2z}\right)^z \cdot \frac{1}{\alpha} \cdot 2^{i+1} \cdot \Delta\}$. Let $R_M = \bigcup_{i=1}^{\log\left(\frac{2z}{\varepsilon}\right)^{3z}} R_i$. For each ring $R_i$ with $1 \leq i \leq \log\left(\frac{2z}{\varepsilon}\right)^{3z}$, we sample a subset $S_i$ of $s$ points uniformly at random, where each point is weighted by $\frac{|R_i|}{s}$. We weigh $q$ by the number of points in $A \setminus R_M$. Finally, we set $\kappa = \sum_{i > \log\left(\frac{2z}{\varepsilon}\right)^{3z}} \sum_{p \in R_i} \|p - q\|^z$. We claim, for $s \in \tilde{O}(\varepsilon^{-2})$ that for the weights defined above, $\bigcup_{i=1}^{\log\left(\frac{2z}{\varepsilon}\right)^{3z}} S_i \cup \{q\}$ together with the constant $\kappa$ is a coreset.

The analysis for every ring $R_i \in R_M$ is merely an application of Theorem 4. For the remaining points, we use the following lemma.

---

[4]We described an option in detail for the sublinear algorithm. In the interest of keeping the presentation succinct, we defer to that part of the paper and omit further discussion.

**Lemma 13.** *Let $q$, $R_M$ and $\kappa$ be defined as above. Then for any point $c \in \mathbb{R}^d$, we have*

$$|cost(A, c) - (cost(R_M, c) + \kappa + |A \setminus R_M| \cdot \|q - c\|^2)| \le \varepsilon \cdot cost(A, c).$$

## 5   Experimental Evaluation

While we can prove that the algorithm can compute a good solution in constant time for every constant $\varepsilon$ and $z$, even for moderately small $\varepsilon$ (e.g. $\varepsilon = 1/2$) the sampling complexity becomes quite large for even small values of $z$, as indeed our lower bound shows is necessarily the case. Our experiments therefore aim at evaluating the performance of the sublinear algorithm on realistic, not necessarily worst case data sets.

As baseline algorithm, we implemented a simple version of a batched gradient descent. Since all considered objectives are convex, we can expect such an algorithm to find a good solution in a reasonable time. The sublinear algorithm ran Algorithm 1 before calling the batched gradient descent. The code can be found at `https://github.com/DaSau/power-mean` We selected two data sets from the UCI repository Dua & Graff (2017), both of which are under the Creative Commons license. The first data set is the 3D Road Network data set from Kaul et al. (2013). It consists of elevation information with the attributes longitude, latitude and altitude. The total number of points is 434,874. For this data set, we considered all powers from $z = 3$ to $7$. The second data set is the USCensus data set, consisting of the records from a 1990 census. The total size of the data set was 2,458,285 samples, each with 68 attributes. For this data set, we considered the powers $z = 3, 4, 5$.

The results essentially confirmed that the sublinear algorithm succeeded in finding a good candidate solution in a fraction of the time as batch gradient descent for essentially all considered problems.

A more extensive discussion can be found in the supplementary material.

## 6   Conclusion and Future Work

We gave sublinear algorithms and coresets for any power of means. Our bounds are nearly tight for the sublinear algorithms and we conjecture the coreset bound to be optimal, up to polylog factors.

The most immediate open question is whether our results generalize to coresets for $k$-clustering objectives. It seems likely that coresets of size $O(k^2/\varepsilon^2)$ are achievable using our techniques. Improving on either this bound or the $O(k\varepsilon^{-2-z})$ from Cohen-Addad et al. (2021) is arguable the most important open problem in coresets.

In terms of sublinear algorithms, there is still a sub-optimal dependency on the $\varepsilon$ by a factor $\varepsilon^{-O(1)}$. Obtaining tight bounds would be interesting. Finally, it is also an interesting open question whether sublinear algorithms for $\ell_p$ with $p > 2$ exist. It is known that for these spaces, no coreset that is independent of $d$ can exist, even for the mean or the median. Is it nevertheless possible to obtain a sublinear algorithm that is independent of $d$?

## Acknowledgments and Disclosure of Funding

The work of David Saulpic is [partially] funded by the grant ANR-19-CE48-0016 from the French National Research Agency (ANR).

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
