## A  Additional Definitions and Properties

**Definition 4.** *Let $X$ be a ground set, and $\mathcal{R} \subset \mathcal{P}(X)$. We say that $(X, \mathcal{R})$ is a range space.*

*The VC-dimension of a range space $(X, \mathcal{R})$ is the largest $d$ such that, for some $S \subseteq X$ with $|S| = d$, $|\{R \cap S \mid R \in \mathcal{R}\}| = 2^d$.*

Let us consider the range space induced by Euclidean balls centered around a single point $p$. A range $R$ is induced by a ball of radius $r$ is the set of all points at distance $r$ or less from $p$, i.e. $R = \{q \in X \mid \|p - q\| \leq r\}$. Without loss of generality, assume that $q$ is the origin. We show that for any set of two points, we cannot generate all possible dichotomies, i.e. for any point set $S$ we

have $|\{R \cap S \mid R \in \mathcal{R}\}| < 4$. If both points have the same distance from $p$, then it is not possible to define a range that contains one point and not the other. If both points have different distance from $p$, it is not possible to define a range that contains the furthest point, but not the closest.

## B  Uniform Sampling Routine

**Lemma 4.** *Let* $\mathbb{N}_{\varepsilon/10}$ *be an* $\varepsilon/10$-*net of* $R$. *Then if* $\sup_{v \in \mathbb{N}_{\varepsilon/10}} \frac{\left|\sum_{j=1}^{s} \frac{|R|}{s} v_{p_j} - \|v\|_1\right|}{cost(R,q)+cost(R,c)} \leq \frac{\varepsilon}{10}$ *we have*
$\sup_{c \in \mathbb{R}^d} \frac{\left|\sum_{j=1}^{s} \frac{|R|}{s} \|p_j - c\|^z - cost(R,c)\right|}{cost(R,q)+cost(R,c)} \leq \varepsilon$ *for all* $c \in \mathbb{R}^d$.

*Proof.* Let $c \in \mathbb{R}^d$ be an arbitrary point. We first deal with the case where there is some point $p' \in R_i$ with $\|p' - c\|^z \geq \left(\frac{8z}{\varepsilon}\right)^z \cdot \|p' - q\|^z$. Then, for any point $p'' \in R$

$$\|p'' - c\|^z$$

$$(Lemma\ 1) \quad \leq \quad (1+\varepsilon) \cdot \|p' - c\|^z + \left(\frac{2z+\varepsilon}{\varepsilon}\right)^{z-1} \cdot \|p' - p''\|^z$$

$$\leq \quad (1+\varepsilon) \cdot \|p' - c\|^z + \left(\frac{2z+\varepsilon}{\varepsilon}\right)^{z-1} \cdot 2^{z+1}\|p' - q\|^z$$

$$\leq \quad (1+\varepsilon) \cdot \|p' - c\|^z + \left(\frac{2z+\varepsilon}{\varepsilon}\right)^{z-1} \cdot 2^{z+1} \cdot \left(\frac{\varepsilon}{8z}\right)^z \cdot \|p' - c\|^z$$

$$\Rightarrow \frac{\|p'' - c\|^z}{\|p' - c\|^z} \quad \leq \quad (1+\varepsilon)$$

Using an analogous calculation, one can also show

$$\frac{\|p' - c\|^z}{\|p'' - c\|^z} \leq (1+\varepsilon)$$

which implies that $(1-\varepsilon) \cdot \|p' - c\|^z \leq \|p'' - c\|^z \leq (1+\varepsilon) \cdot \|p' - c\|^z$, ..e., any point in $R$ costs the same up to a $(1 \pm \varepsilon)$ factor. Since the coreset weights, by construction, sum up to $|R|$, we therefore have

$$|cost(R,c) - cost(\Omega,c)| \leq \varepsilon \cdot cost(R,c) \tag{2}$$

Now, we focus on the case where $\|p - c\|^z \leq \left(\frac{8z}{\varepsilon}\right)^z \cdot \|p - q\|^z$ for all $p \in R$. We will assume

$$\sup_{v \in N_\varepsilon} \frac{\left|\sum_{j=1}^{s} \frac{|R|}{s} v_{p_j} - \|v\|_1\right|}{cost(R,q) + cost(R,c)} \leq \varepsilon$$

and rescale $\varepsilon$ by a factor of 10 at the end. We know that there exists net vector $v \in N_\varepsilon$ such that

$$\left|\|p - c\|^z - v_p\right| \leq \varepsilon \cdot \left(\|p - c\|^z + \|p - q\|^z\right).$$

Summing this over all points in $R$, we therefore have

$$\left|\sum_{p \in R} \left(\|p - c\|^z - v_p\right)\right| \quad \leq \quad \sum_{p \in R} \left|\|p - c\|^z - v_p\right|$$

$$\leq \quad \varepsilon \cdot \sum_{p \in R} \|p - c\|^z + \|p - q\|^z$$

$$\leq \quad \varepsilon \cdot (cost(R,c) + cost(R,q)). \tag{3}$$

We similarly show that $\sum_{j=1}^{s} \frac{|R|}{s} \|p_j - c\|^z$ and $v_{p_j}$ are close. First observe that if $\left|\sum_{p \in R} v_p - \sum_{j=1}^{s} \frac{|R|}{s} v_{p_j}\right| \leq \varepsilon \cdot \sum_{p \in R} v_p$, as assumed in the lemma, then $\sum_{j=1}^{s} \frac{|R|}{s} v_{p_j} \leq$

$2\left|\sum_{p\in R} v_p\right|$. Therefore we have

$$
\begin{aligned}
\left|\sum_{j=1}^{s} \frac{|R|}{s} \cdot \left(\|p_j - c\|^z - v_{p_j}\right)\right| &\leq \sum_{j=1}^{s} \frac{|R|}{s} \cdot \left|\|p_j - c\|^z - v_{p_j}\right| \\
&\leq \varepsilon \cdot \sum_{j=1}^{s} \frac{|R|}{s} \left(\|p_j - c\|^z + \|p_j - q\|^z\right) \\
&\leq 2\varepsilon \cdot \sum_{j=1}^{s} \frac{|R|}{s} \left(v_{p_j} + 2 \cdot \frac{\text{cost}(R, q)}{|R|}\right) \\
&\leq 2\varepsilon \cdot \left(2\text{cost}(R, q) + \sum_{p\in R} v_p\right) \\
&\leq 4\varepsilon \cdot \left(\text{cost}(R, c) + \text{cost}(R, q)\right). \quad (4)
\end{aligned}
$$

Combining equations 3 and 4, we therefore obtain

$$
\begin{aligned}
\left|\sum_{p\in R} \|p - c\|^z - \sum_{j=1}^{s} \frac{|R|}{s} \cdot \|p_j - c\|^z\right| &\leq \left|\sum_{p\in R} \|p - c\|^z - v_p\right| \\
&+ \left|\sum_{j=1}^{s} \frac{|R|}{s} \cdot \|p_j - c\|^z - \sum_{j=1}^{s} \frac{|R|}{s} \cdot v_{p_j}\right| \\
&+ \left|\sum_{p\in R} v_p - \sum_{j=1}^{s} \frac{|R|}{s} \cdot v_{p_j}\right| \\
&\leq 10\varepsilon \cdot \left(\text{cost}(R, c) + \text{cost}(R, q)\right).
\end{aligned}
$$

Together with Equation 2 and rescaling $\varepsilon$ by a factor of 10 yields the claim. $\qquad\square$

**Lemma 5.** *Let $R$ and $q$ be defined as above. Then for every $\beta > 0$, there exists a $\beta$-net of $R$ of size at most $\exp(\gamma \cdot z^3 \beta^2 \log \|R\|_0 \cdot \log \varepsilon^{-1})$, where $\gamma$ is an absolute constant.*

*Proof.* Let $f : \mathbb{R}^d \to \mathbb{R}^k$ with $k \in O(z^2 \beta^{-2} \log R_i)$ be a terminal embedding satisfying for all $c \in \mathbb{R}^d$ and all $p \in R_i$

$$
(1 - \beta/2z) \cdot \|p - c\| \leq \|f(p) - f(c)\| \leq (1 + \beta/2z) \cdot \|p - c\|.
$$

Note that this also implies

$$
(1 - \beta) \cdot \|p - c\|^z \leq \|f(p) - f(c)\|^z \leq (1 + \beta) \cdot \|p - c\|^z.
$$

We now discretise $\mathbb{R}^k$ as follows. We cover the entire $k$-sphere centred around $q$ with radius $\left(\frac{8z}{\varepsilon}\right) \cdot 2 \cdot \Delta_i^{\frac{1}{z}}$ with $k$-dimensional balls of radius at most $\frac{\varepsilon}{3z} \cdot \Delta_i^{\frac{1}{z}}$. Let $B$ be the minimal set of balls required. In $k$-dimensional Euclidean spaces, such a cover has size at most

$$
\left(1 + 2 \cdot \left(\frac{8z}{\varepsilon}\right)^z / \varepsilon\right)^k.
$$

For every center $c'$ of some ball in $B$, we add the vector $v'$ with entries $v'_p = \|f(p) - c'\|^z$ to the net $N$. We claim that $N$ is an $O(\beta)$-net. The lemma then follows by rescaling $\beta$.

For every point $c$ with $\|p - c\| \leq \frac{8z}{\varepsilon} \cdot \|p - q\|$, let $f(c)$ be the image of this point under the terminal embedding. Moreover, let $v'$ be the vector induced by the point $c'$ in $B$ closest to $f(c)$. We have

$$
\begin{aligned}
& |\|p - c\|^z - v_p| \\
= \quad & |\|p - c\|^z - \|f(p) - f(c)\|^z + \|f(p) - f(c)\|^z - v_p| \\
= \quad & |\|p - c\|^z - \|f(p) - f(c)\|^z + \|f(p) - f(c)\|^z - v_p| \\
(Lemma\ 1) \quad \leq \quad & \beta \cdot \|p - c\|^z + \varepsilon \cdot \|f(p) - f(c)\|^z + \left(\frac{2z+1}{\varepsilon}\right)^{z-1} \|f(c) - c'\|^z \\
\leq \quad & 3\beta \|f(p) - f(c)\|^z + \left(\frac{3z}{\varepsilon}\right)^{z-1} \cdot \left(\frac{\varepsilon}{3z}\right)^z \cdot \Delta_i \\
\leq \quad & 3\beta \|f(p) - f(c)\|^z + \varepsilon \|p - q\|^z \\
\leq \quad & 3\beta (\|f(p) - f(c)\|^z + \|p - q\|^z).
\end{aligned}
$$

Rescaling $\beta$ completes the proof. $\qquad\square$

**Lemma 6.** *Let $R$ and $q$ be defined as above and let $\Omega$ be a uniform sample consisting of $s$ points. Then for any point $c$ and $s \geq \eta \cdot z^3 2^{8z} \cdot \varepsilon^{-2} \cdot \log \|R\|_0 \cdot \log^3 \varepsilon^{-1}$ for some absolute constant $\eta$, we have*
$$
\mathbb{E}_\Omega \mathbb{E}_g \sup_{c \in \mathbb{R}^d} \left| \frac{\sum_{j=1}^s \frac{|R|}{s} \|p_j - c\|^z \cdot g_j}{cost(R,q) + cost(R,c)} \right| \leq \varepsilon. \ \text{Moreover, if } s \geq \eta \cdot z^3 2^{8z} \cdot \varepsilon^{-2} \cdot \log \|R\|_0 \cdot \log^4 \varepsilon^{-1} \log 1/\delta
$$

*for some absolute constant $\eta$, then we have* $\sup_{c \in \mathbb{R}^d} \left| \frac{\sum_{j=1}^s \frac{|R|}{s} \|p_j - c\|^z \cdot g_j}{cost(R,q) + cost(R,c)} \right| \leq \varepsilon$, *with probability at least* $1 - \delta$.

*Proof.* Lemma 4 shows that in order to bound $\mathbb{E}_\Omega \mathbb{E}_g \sup_{c \in \mathbb{R}^d} \left| \frac{\sum_{j=1}^s \frac{|R|}{s} \|p_j - c\|^z \cdot g_j}{cost(R,q) + cost(R,c)} \right|$, it is enough to take the supremum only over vectors of $N_{\varepsilon/10}$.

For a center $c \in \mathbb{R}^d$, let $v$ be the cost vector induced by $c$ and let therefore $v' \in N_{\varepsilon/10}$ be the approximation of $v$ given by the net. Furthermore, let $v'^0 = 0, v'^1, v'^2, \ldots$ be a sequence of cost vectors such that $v' = \sum_{k=1}^{\log \varepsilon/10} v'^{k+1} - v'^k$ and $v'^k \in \mathbb{N}_{2^{-k}}$.

$$
\begin{aligned}
& \mathbb{E}_\Omega \mathbb{E}_g \sup_{v' \in N_{\varepsilon/10}} \left| \frac{\sum_{j=1}^s \frac{|R|}{s} v'_{p_j} \cdot g_j}{cost(R,q) + cost(R,c)} \right| \\
= \quad & \mathbb{E}_\Omega \mathbb{E}_g \sup_{v' \in N_{\varepsilon/10}} \sum_{k=1}^{\log 10\varepsilon^{-1}} \left| \frac{\sum_{j=1}^s \frac{|R|}{s} (v'^{k+1}_{p_j} - v'^k_{p_j}) \cdot g_j}{cost(R,q) + cost(R,c)} \right| \\
\leq \quad & \mathbb{E}_\Omega \sum_{k=1}^{\log \varepsilon/10} \mathbb{E}_g E'^{k+1}
\end{aligned}
$$

with $E'^{k+1} = \sup_{v'^{i+1}, v'^i \in \mathbb{N}_{k+1} \times \mathbb{N}_k} \left| \frac{\sum_{j=1}^s \frac{|R|}{s} (v'^{i+1}_{p_j} - v'^i_{p_j}) \cdot g_j}{cost(R,q) + cost(R,c)} \right|$ so we now focus on bounding the supre-

mum over the $E'^{k+1}$. For every $i$, $\sum_{j=1}^s \frac{|R|}{s} \frac{(v'^{i+1}_{p_j} - v'^i_{p_j}) \cdot g_j}{cost(R,q) + cost(R,c)}$ is Gaussian distributed with zero mean and variance

$$
\begin{aligned}
& \sum_{j=1}^s \left( \frac{|R|}{s} \frac{(v'^{i+1}_{p_j} - v'^i_{p_j})}{cost(R,q) + cost(R,c)} \right)^2 \\
\leq \quad & \sum_{j=1}^s \left( \frac{|R|}{s} \frac{(v'^{i+1}_{p_j} - v_{p_j} + v_{p_j} - v'^i_{p_j})}{cost(R,q) + cost(R,c)} \right)^2 \\
\leq \quad & \sum_{j=1}^s \frac{|R|^2}{s^2} \cdot \frac{2^{-2k} \cdot 2 \cdot v_{p_j}^2}{(cost(R,q) + cost(R,c))^2}
\end{aligned}
$$

We distinguish between two cases. If $v_{p_j} = \text{cost}(p_j, c) \geq 8^z \cdot \text{cost}(p_j, q)$, then for any point $p'$ $\text{dist}(p, c) \leq \text{dist}(p', c) + \text{dist}(p_j, p') \leq \text{dist}(p', c) + 3\text{dist}(p_j, q) \leq \text{dist}(p', c) + \frac{3}{8}\text{dist}(p_j, c)$, and hence $\text{cost}(p_j, c) \leq 8^z\text{cost}(p', c)$, and by averaging $v_{p_j} \leq 8^z \cdot \frac{\text{cost}(R,c)}{|R|}$. Otherwise $v_{p_j} = \text{cost}(p, c) \leq 8^z\text{cost}(p, q) \leq 2 \cdot 8^z \cdot \frac{\text{cost}(R,q)}{|R|}$. Combining both in the aforementioned variance bound, we have:

$$\frac{1}{s} \sum_{j=1}^{s} \frac{2^{-2k} \cdot 2 \cdot v_{p_j}^2}{(\text{cost}(R,q) + \text{cost}(R,c))^2} \cdot \frac{|R|^2}{s}$$

$$\leq \frac{1}{s} \sum_{j=1}^{s} \frac{2^{-2k+3} \cdot 8^{2z} \cdot \max\left(\frac{\text{cost}(R,c)}{|R|}, \frac{\text{cost}(R,q)}{|R|}\right)^2}{(\text{cost}(R,q) + \text{cost}(R,c))^2} \cdot \frac{|R|^2}{s}$$

$$\leq \frac{1}{s} 8^{2z+1} \cdot 2^{-2k}$$

Now since $E_k$ is the supremum of at most $|\mathbb{N}_{k+1}| \cdot |\mathbb{N}_k|$ many Gaussians, and the expected maximum over $n$ Gaussians with varaince at most $\sigma^2$ is at most $\sqrt{2 \log n \cdot \sigma^2}$, we have

$$\mathbb{E}_g \sup E'^{k+1} \leq \sqrt{2 \log \left(|\mathbb{N}_{k+1}|^2\right) \cdot \frac{1}{s} \cdot 8^{2z+1} \cdot 2^{-2k}}$$

$$\leq \sqrt{2\gamma \cdot z^3 \cdot 2^{2k} \log |R| \cdot \log \varepsilon^{-1} \cdot \frac{1}{s} \cdot 8^{2z+1} 2^{-2k}}$$

$$\leq \varepsilon / \log 10\varepsilon^{-1},$$

where the first inequality follows from Lemma 5 and the second inequality holds by our choice of $s$. Therefore, $\mathbb{E}_g E'^{k+1} \leq \varepsilon / \log 1/\varepsilon$ and consequently

$$\mathbb{E}_\Omega \mathbb{E}_g \sup_{v' \in N_{\varepsilon/10}} \sum_{k=1}^{\log 10\varepsilon^{-1}} \left| \frac{\sum_{j=1}^{s} \frac{|R|}{s} v'_{p_j} \cdot g_j}{\text{cost}(R,q) + \text{cost}(R,c)} \right|$$

$$\leq \mathbb{E}_\Omega \sum_{k=1}^{\log 10\varepsilon^{-1}} \mathbb{E}_g E'^{k+1}$$

$$\leq \log 10\varepsilon^{-1} \cdot \varepsilon / \log 10\varepsilon^{-1} \leq \varepsilon.$$

The proof now follows since Lemma 4 states that is is sufficient to get a bound for all cost vectors in $N_{\varepsilon/10}$ in order to get a coreset for all $c \in \mathbb{R}^d$, up to a rescaling of $\varepsilon$ by a factor 10.

To obtain a high probability bound, we now merely observe that for a zero mean Gaussian $g$ with variance $\sigma^2$, we have $\mathbb{P}[g > t] \leq \frac{1}{2\pi} \exp(-t^2/(2\sigma^2))$. Hence, we can simply take a union bound over all steps of the chain an obtain

$$\mathbb{P}[\exists v'^{i+1}, v'^i \in \mathbb{N}_{i+1} \times \mathbb{N}_i \mid E_i > \varepsilon / \log 10\varepsilon^{-1}]$$

$$\leq |\mathbb{N}_{i+1}| \cdot |\mathbb{N}_i| \cdot \frac{1}{2\pi} \cdot \exp\left(\frac{\varepsilon^2}{\log^2 10\varepsilon^{-1}} \cdot \frac{8^{2z+2} \cdot 2^{-2k}}{s}\right).$$

The claim now follows by the second choice of $s$, and taking a union bound over all elements of the chain. $\qquad\square$

**Lemma 7.** *Let $R$ and $q$ be defined as above. Suppose a uniform sample of size $s \in \tilde{O}(\Gamma \cdot \log \|R\|_0 \cdot \log 1/\delta)$ satisfies with probability at least $1 - \delta$ for all candidate centers $c$*

$$\left| cost(R, c) - \sum_{p \in \Omega} \frac{|R|}{s} \cdot \|p - c\|^z \right| \leq \varepsilon \cdot (cost(R, q) + cost(R, c)).$$

*Then a uniform sample of size $\tilde{O}(\Gamma \cdot \log 1/\delta)$ achieves the same guarantee.*

*Proof.* We start by briefly outlining the key arguments from Theorem 3.1 of Braverman, Jiang, Krauthgamer, and Wu Braverman et al. (2021). Denote by $\log^{(i)} n$ the $i$-fold logarithm of $n$, i.e. $\log^{(2)} n = \log \log n$. Suppose that the initial summary has size $\Gamma \cdot \log \|R\|_0$. We call the sampling algorithm recursively. In iteration $i$, let $\|R_i\|_0$ be the distinct number of points left in iteration $i$, let $\varepsilon_i$ be the precision parameter used in iteration $i$ and let $\delta_i$ be the failure probability. We choose the parameters $\varepsilon_i := \varepsilon/(\log^{(i)} \|R\|_0)^{\frac{1}{2}}$ and $\delta_i := \delta/\|R_{i-1}\|_0$.

BJKW show the invariants $\|R_i\|_0 \leq 20\Gamma \log \delta^{-1} (\log^{(i)} \|R\|_0)^3$, $\prod_{i=1}^{t}(1+\varepsilon_i) \leq \exp(2\varepsilon_t)$ and $\sum_{i=1}^{t} \delta_i \leq \delta \cdot \left( \frac{1}{\|R\|_0} + \frac{1}{\log \|R\|_0} + \cdots + \frac{1}{\log^{(t-1)} \|R\|_0} \right) \in O(\delta)$. The first invariant shows that after $O(\Gamma^4)$ iterations, $\log^{(t)} \|R\|_0 \leq 20\Gamma$. The second invariants bounds the overall error, which, by choice of $\varepsilon_t$ and the maximum number of iterations, is less than $(1 + O(\varepsilon))$. The final invariant shows that the overall failure increases only by constant factors.

We now argue why it is sufficient for an algorithm that only uses uniform sampling to target the final bound after the iterative size reduction. Consider the sampling distributions $D_0$ and $D_t$, where $D_0$ is the sampling distribution before and and $D_t$ is the sampling distribution after the iterative size reduction. Note that since uniform sampling assigns the exact same weight to every point, the sampling itself remains weight oblivious.

Therefore the sampling distribution of every application of Lemma 3 remains uniform sampling, i.e. the probability that a point $p$ is in the output of $D_t$ is equal for all points. The same holds for the output of $D_0$, therefore the distributions of both algorithms are identical. Since $D_t$ achieves a the desired guarantee by sampling $\tilde{O}(\varepsilon^{-2} \cdot 2^{O(z)})$ many distinct points, $D_0$ must do so as well. $\qquad\square$

## C   Pruning Lemmas

**Lemma 8.** *Suppose we are given an $\alpha$-approximate center $q$. Let $B(q,r)$ be the ball centered at $q$ with radius $r = 4 \cdot \left( \frac{2\alpha \cdot OPT}{n} \right)^{1/z}$. Then the following two statements hold.*

   1. *Any $\alpha$-approximate center is in $B(q,r)$.*

   2. *For any two points $c, c' \in B(q,r)$ and for any point $p$ with $\|p - q\|^z > \gamma \cdot r^z$ with $\gamma > \varepsilon^{-z} \cdot (12z)^z$, we have $\|p - c\|^z \leq (1 + \varepsilon) \cdot \|p - c'\|^z$.*

*Proof.* For the first claim we consider a point $c$ not in $B(q,r)$ and show that $c$ cannot be an $\alpha$-approximate center.

The average cost of the points when using $q$ as a center is $\alpha \cdot \frac{OPT}{n}$. Hence, by Markov's inequality, at least half of the points of $A$ lie in $B(q, r/4)$. Furthermore, by choice of $c$ and the triangle inequality, we have $\|p - c\| > 2 \cdot \|p - q\|$ for any point $p \in B(q, r/4)$. Hence, the cost of clustering all the points in $A \cap B(q, r/4)$ to $c$ is at least $n/2 \cdot (2 \cdot r)^z \geq \alpha \cdot OPT$.

For the second claim, let $c, c' \in B(q,r)$ and $p$ with $\|p - q\|^z \geq \gamma \cdot r^z$. We first note that $\|p - c'\| \geq \|p - q\| - \|q - c'\| \geq \gamma^{1/z} \cdot r - 2r = (\gamma^{1/z} - 2)r$, which yields the inequality

$$r \leq \|p - c'\| \cdot \frac{1}{\gamma^{1/z} - 2} \tag{5}$$

We then have

$$\|p - c\|^z$$

$$(Lem.\ 1) \quad \leq \quad (1 + \varepsilon/2z)^{z-1} \|p - c'\|^z + \left(\frac{\varepsilon + 2z}{\varepsilon}\right)^{z-1} \cdot \|c - c'\|^z$$

$$\leq \quad (1 + \varepsilon/2) \cdot \|p - c'\|^z + \left(\frac{3z}{\varepsilon}\right)^{z-1} (2r)^z$$

$$(Eq.\ 5) \quad \leq \quad (1 + \varepsilon/2) \cdot \|p - c'\|^z + \left(\frac{3z}{\varepsilon}\right)^{z-1} 2^z \cdot \|p - c'\|^z \left(\frac{1}{\gamma^{1/z} - 2}\right)^z$$

$$\leq \quad (1 + \varepsilon/2) \cdot \|p - c'\|^z + \left(\frac{3z}{\varepsilon}\right)^{z-1} 4^z \cdot \|p - c'\|^z \cdot \gamma^{-1}$$

$$(\text{Choice of } \gamma) \quad \leq \quad (1 + \varepsilon/2) \cdot \|p - c'\|^z + \varepsilon/2 \cdot \|p - c'\|^z$$

$$\leq \quad (1 + \varepsilon) \cdot \|p - c'\|^z$$

$\square$

**Lemma 9.** *Suppose that $q$ is an $\alpha$-approximate solution. Let $A_{near} \subset A$ be a set of points with cost at most $(\varepsilon/(\alpha 5z))^z \cdot \frac{OPT}{n}$. Let $\hat{A} = (1 \pm \varepsilon)|A_{near}|$ Then for any candidate solution $c$ we have*

$$\left| \hat{A} \cdot \|q - c\|^z - \sum_{p \in A_{near}} \|p - c\|^z \right| \leq \varepsilon/\alpha \cdot \left( \sum_{p \in A_{near}} \|p - c\|^z + OPT \right).$$

*Proof.* We first prove the result for $\hat{A} = |A_{near}|$, the claim for an estimation of $|A_{near}|$ is a simple corollary. We have, using Lemma 1:

$$\left| \sum_{p \in A_{near}} (\|p - c\|^z - \|q - c\|^z) \right|$$

$$\leq \quad \sum_{p \in A_{near}} |\|p - c\|^z - \|q - c\|^z|$$

$$\leq \quad \sum_{p \in A_{near}} \left( \frac{\varepsilon}{\alpha 2} \cdot \|p - c\|^z + \left(\frac{\alpha 5z}{\varepsilon}\right)^{z-1} \|p - q\|^z \right)$$

$$\leq \quad \sum_{p \in A_{near}} \left( \frac{\varepsilon}{\alpha 2} \cdot \|p - c\|^z + \left(\frac{\alpha 5z}{\varepsilon}\right)^{z-1} \left(\frac{\varepsilon}{\alpha 5z}\right)^z \frac{OPT}{n} \right)$$

$$\leq \quad \varepsilon/\alpha \cdot \left( \sum_{p \in A_{near}} \|p - c\|^z + \text{OPT} \right)$$

For an approximation to $|A_{near}|$ we now merely add an additional additive error $\varepsilon \cdot \sum_{p \in A_{near}} \|p - c\|^z$ to the difference of the two terms. $\square$

**Lemma 10.** *Suppose that $q$ that is an $\alpha$-approximate solution. Let $R_{cheap}$ the union of rings with $|R_i \cap A| < \varepsilon \cdot \eta \cdot n$ and with radius at most $4(\gamma \cdot \frac{\alpha \cdot OPT}{n})^{1/z}$, where $\gamma$ is given by Lemma 8. Then, for any candidate solution $c$*

$$\sum_{p \in R_{cheap} \cap A} \|p - c\|^z \leq \varepsilon \cdot 4^{z+1} \cdot \beta \cdot \eta \cdot \gamma \cdot \alpha \cdot \sum_{p \in A} \|p - c\|^z.$$

*Proof.* We first require a bound on $\|q - c\|^z$. Using Lemma 1, we have

$$n \cdot \|q - c\|^z \leq 2^z \sum_{p \in A} \|p - q\|^z + \|p - c\|^z$$

$$\leq \alpha \cdot 2^{z+1} \cdot \sum_{p \in A} \|p - c\|^z$$

$$\Rightarrow \|p - c\|^z \leq \alpha \cdot 2^{z+1} \cdot \frac{\sum_{p \in A} \|p - c\|^z}{n} \tag{6}$$

Therefore with another application of Lemma 1

$$\sum_{p \in R_{cheap} \cap A} \|p - c\|^z$$

$$\leq \sum_{p \in R_{cheap} \cap A} 2^z \cdot (\|p - q\|^z + \|q - c\|^z)$$

$$(Eq.\ 6) \leq \beta \cdot \varepsilon \cdot \eta \cdot n \cdot 2^z \cdot \left( \gamma \cdot \frac{\alpha \cdot OPT}{n} \right.$$

$$\left. + \alpha \cdot 2^{z+1} \cdot \frac{\sum_{p \in A} \|p - c\|^z}{n} \right)$$

$$\leq \beta \cdot \varepsilon \cdot \eta \cdot \alpha \cdot \gamma 4^{z+1} \cdot \sum_{p \in A} \|p - c\|^z.$$

$\square$

**Lemma 11.** *Let $q$ be a point that is an $\alpha$-approximation and let $S$ be a uniform sample consisting of $O\left(\alpha \cdot \eta^{-1} \cdot \varepsilon^{-3} polylog(\varepsilon^{-1} \cdot \delta^{-1})\right)$ points. Then with probability at least $1 - \delta$ for all rings $R_i$,*

$$|R_i \cap A| - \varepsilon \cdot \max\left(n \cdot \eta, |R_i \cap A|\right) \leq \frac{|R_i \cap S| \cdot n}{|S|} \leq |R_i \cap A| + \varepsilon \cdot \max\left(n \cdot \eta,\ |R_i \cap A|\right).$$

*Furthermore, let $d$ as in the algorithm, i.e., such that $\frac{2}{3} \cdot \varepsilon \cdot \eta \cdot |S| \leq \left|S \setminus (B(q, d) \cap S)\right|$. Then $d < \left(3(\varepsilon\eta)^{-1} \cdot \frac{\alpha \cdot OPT}{n}\right)^{1/z}$*

*Proof.* We consider the range space induced by Euclidean balls centered around $q$. This range space has VC dimension of exactly 2. The VC dimension induced by the intersection of two Euclidean balls centered around $q$ is still constant Blumer et al. (1989), hence for our choice of $|S|$, Lemma 2 ensures that we have approximated the cardinality of all rings up to the additive error $\varepsilon \cdot \max(\eta \cdot n, |R_i \cap A|)$ with probability at least $1 - \delta$, which proves the first claim.

For the second claim, let $d$ as in the algorithm. By Lemma 2, we have $\left|S \setminus (B(q, d) \cap S)\right| \cdot \frac{n}{|S|} \leq (1 + \varepsilon)\left|A \setminus (B(q, d) \cap A)\right|$. Hence, we have

$$\left|A \setminus (B(q, d) \cap A)\right| \geq \frac{2}{3} \cdot \varepsilon \cdot \eta \cdot |S| \cdot \frac{n}{|S|(1 + \varepsilon)} \geq \frac{\varepsilon \cdot \eta \cdot n}{3}$$

Using Markov's inequality, we now know that the number of points with cost $3(\varepsilon\eta)^{-1} \cdot \frac{\alpha \cdot OPT}{n}$ is at most $\frac{\varepsilon \cdot \eta \cdot n}{3}$. This implies $d \leq \left(3(\varepsilon\eta)^{-1} \cdot \frac{\alpha \cdot OPT}{n}\right)^{1/z}$. $\square$

## D  Proof of Theorem 1

We start by specifying our parameters: the approximation is set to be $\alpha = 20^z$. To prune the far points, we set $\gamma = (12z/\varepsilon)^z$. Finally, we pick $\beta$ and $\eta$ such that $\eta = \frac{1}{2^{z-1}\alpha \cdot \beta \cdot \gamma}$ and $3 \cdot 2^{-z\beta+z} \cdot \frac{\alpha}{\varepsilon\eta} \leq \left(\frac{\varepsilon}{5\alpha}\right)^z$. This is possible for $\beta = O_z(\log(1/\varepsilon))$ and $\eta \in O_z(\varepsilon^{-z}polylog(1/\varepsilon))$

First, note that Lemma 8 shows that it is enough to compute an approximate solution for the set $A'$ consisting of points that are at distance less than $O\left(\left(\gamma \frac{OPT}{n}\right)^{1/z}\right)$.

Let $c$ be the optimal $(1, z)$-center. First, let us consider the initial sampled point $q$. With probability at least $9/10$, we have $\|q - c\|^z \leq 10 \cdot \frac{OPT}{n}$. Hence due to the triangle inequality $\sum_{p \in A} \|p - q\|^z \leq 20^z \cdot OPT$, and $q$ is an $\alpha$-approximation. Let $S$ be the set of points sampled and pruned by the algorithm.

For each ring $R_i$ at distance $(2^{-i}d, 2^{-i+1}d]$, we denote by $\hat{R}_i$ either a $(1 \pm \varepsilon)$-estimate of the size of $|R_i|$ via $\frac{n \cdot |R_i \cap S|}{|S|}$, if $|R_i \cap S| \geq \varepsilon \cdot \eta \cdot |S|$, or we set $\hat{R}_i = 0$ if $|R_i \cap S| < \varepsilon \cdot \eta \cdot |S|$. Similarly, define $R_\beta$ to be a $(1 \pm \varepsilon)$-estimate of the size of points at distance less than $2^{-\beta}d$ from $q$, if $R_\beta \geq \varepsilon \cdot \eta \cdot |S|$, or 0 if $R_\beta < \varepsilon \cdot \eta \cdot |S|$.

Our goal to show that for any candidate solution $c'$, we have

$$\left| \sum_{p \in A'} \|p - c'\|^z - \left( \sum_{i \leq \beta - 1} \frac{\hat{R}_i}{|R_i \cap S|} \sum_{p \in R_i \cap S} \|p - c'\|^z + \frac{\hat{R}_\beta}{|R_\beta \cap S|} \|q - c'\|^z \right) \right| \leq \varepsilon \cdot \sum_{p \in A'} \|p - c'\|^z.$$
(7)

Hence, computing a $(1 + \varepsilon)$-approximate solution on the set $S$ will give $(1 + \varepsilon)$-approximate solution for $A'$, which is also one for $A$ following Lemma 8.

We now consider all rings $R_i$ centered around $q$ with radius $(2^{-i}d, 2^{-i+1}d]$. First, for $i = \beta \in O(\log 1/\varepsilon)$, we have using Lemma 11 that the cost of points in $R_i$ is, by choice of $\beta$, $(2^{-\beta+1}d)^z \leq 2^{-z\beta+z} \frac{3}{\varepsilon \eta} \frac{OPT}{n} \leq (\varepsilon/(\alpha 5 z))^z \frac{OPT}{n}$, hence we can use Lemma 9 to bound

$$\left| \sum_{p \in A' \cap R_\beta} \|p - c'\|^z - \hat{R}_\beta \|q - c'\|^z \right| \leq \varepsilon/\alpha \cdot \sum_{p \in A_{near}} \|p - c'\|^z + \|p - q\|^z$$

$$\leq \varepsilon \cdot \sum_{p \in A_{near}} \|p - c\|^z + \|p - c'\|^z.$$

Having dealt with the points close to $q$, we now deal with those far away. Since $A'$ results in the pruning of $A$, rings with radius more than $4(\gamma \cdot \frac{\alpha \cdot OPT}{n})^{1/z}$ are empty in $A'$. Moreover, due to Lemma 11, those rings must have $\hat{R}_i = 0$; hence, their contribution to Eq. (7) is 0.

We now turn our attention to the remaining rings. First, we consider the cheap rings, i.e. all rings with $\hat{R}_i = 0$. Note that, by choice of $d$; this includes all rings with $i \leq 1$. We have, due to Lemma 10:

$$\sum_{p \in R_{cheap} \cap A} \|p - c\|^z \leq 4^{z+1} \beta \cdot \varepsilon \cdot \eta \cdot \alpha \cdot \gamma \cdot \left( \sum_{p \in A} \|p - c\|^z \right) \leq \varepsilon \left( \sum_{p \in A} \|p - c\|^z \right)$$

Recall that $\hat{R}_i = 0$ in this case. We therefore obtain

$$\left| \sum_{p \in R_{cheap} \cap A} \|p - c\|^z - \frac{\hat{R}_i}{|R_i \cap S|} \sum_{p \in R_i \cap S} \|p - c\|^z \right| \leq \varepsilon \sum_{p \in A} \|p - c\|^z.$$

Finally, we consider rings with $|R_i \cap S| > \varepsilon \eta |S|$, with $\beta - 1 \leq i \leq 1$. With our choice of $|S| = \frac{\alpha \cdot \text{polylog}(\varepsilon^{-1}\delta^{-1})}{\eta \cdot \varepsilon^3}$ and Lemma 11, we have therefore $|R_i \cap S| > \varepsilon^{-4}\text{polylog}(\varepsilon^{-1})$. Theorem 4 guarantee us that with $\tilde{O}(\varepsilon^{-2})$ many samples, we have

$$\left| \sum_{p \in R_i} \|p - c\|^z - \frac{|R_i|}{m} \sum_{p \in S} \|p - c\|^z \right| \leq \varepsilon/\beta \cdot \left( \sum_{p \in R_i} \|p - c\|^z + \|p - q\|^z \right)$$

Summing up the error for all rings yields a total error of $O(\varepsilon) \cdot \sum_{p \in A'} \|p - c\|^z + \|p - c'\|^z \in O(\varepsilon) \sum_{p \in A'} \|p - c'\|^z$.

Subsequently, we can use any desired optimization algorithm to compute a $(1 + \varepsilon)$-approximate solution $c'$ on $S$, with weights $\frac{\hat{R}_i}{|R_i \cap S|}$ on points of $R_i$. Rescaling $\varepsilon$ according to degree of precision of the optimization procedure and the precision of the coreset completes the proof.

# E  Probability Amplification

**Lemma 12.** *Given query access to A, we can identify with probability $1 - \delta$ a $8^z$-approximate solution using $O(\varepsilon^{-2}(\log 1/\varepsilon + \log 1/\delta) \log 1/\delta)$ samples.*

*Proof.* Let $c$ be the optimal center. In the following we will assume no knowledge of $OPT$, or even an estimate of $OPT$, but we assume to know $n$.

With probability at least $1/2$, a random point $q_i$ satisfies

$$\|q_i - c\|^z \leq 2^z \frac{OPT}{n}.$$

Therefore, when sampling $\log 1/\delta$ points, we will have sampled a $2^z$ approximate solution with probability at least $1 - \delta$.

Furthermore

$$\sum_{p \in A} \|p - q_i\|^z \leq \sum_{p \in A} 2^{z-1} \cdot (\|p - c\|^z + \|q_i - c\|^z) \leq 2^z \cdot OPT.$$

Now since the range space induced by unit Euclidean balls centered around $q$ has VC dimension 2, we can estimate the number of points for any given radius up to an additive error of $\varepsilon \cdot n$. Hence, with probability $1 - \delta$, for every $2^z$-approximate solution $q_i$, the estimated number of points in $B(q, 2\left(4\frac{OPT}{n}\right)^{1/z})$ will be at least $n/2$.

Conversely, if $q_j$ is not $8^z$ approximate, the estimated number of points in $B(q_j, 2\left(4\frac{OPT}{n}\right)^{1/z})$ is small. We have

$$\sum_p \|p - q_j\|^z > 8^z \sum_p \|p - c\|^z$$

$$\Rightarrow \left(\sum_p \|p - q_j\|^z\right)^{1/z} > 8 \left(\sum_p \|p - c\|^z\right)^{1/z}$$

$$\Rightarrow \left(\sum_p \|c - q_j\|^z\right)^{1/z} > 7 \left(\sum_p \|p - c\|^z\right)^{1/z}$$

$$\Rightarrow \|c - q_j\| > 7 \left(\frac{OPT}{n}\right)^{1/z}$$

Therefore, the intersection of $B(q_j, 2\left(4\frac{OPT}{n}\right)^{1/z})$ with $B(c, \left(2\frac{OPT}{n}\right)^{1/z})$ is empty. Since at least $\frac{3n}{4}$ points lie in $B(c, 2\left(4\frac{OPT}{n}\right)^{1/z})$, we know that with probability at least $1 - \delta$ the estimated number of points in $B(q_i, 3\left(\frac{OPT}{n}\right)^{1/z})$ will be larger than the estimated number of points in $B(q_j, 2\left(4\frac{OPT}{n}\right)^{1/z})$. Conditioned on having a $2^z$ approximate solution in our sample, the returned point is therefore no worse than a $8^z$ approximation. $\square$

# F  Lower bound

The goal of that section is to prove Theorem 3, i.e., that any $(1 + \varepsilon)$-approximation algorithm must sample $\varepsilon^{-z+1}$ points.

*Proof of Theorem 3.* Consider the instance $\mathcal{I}$ on the 1-dimensional line where $n$ points are located at 0 and $\varepsilon^{z-1}n$ points are located at 1. Intuitively, we show that any approximation algorithm on $\mathcal{I}$ must sample at least a point at 1, and so must sample at least $\varepsilon^{-z+1}n$ points.

For simplicity, we rescale the instance so that $n = 1$. The optimal solution is $\text{OPT}_{\mathcal{I}} = \inf x^z + \varepsilon^{z-1}(1 - x)^z$, and the optimal center is such that the derivative of the objective function is zero:

$$\frac{\partial}{\partial x} \left(x^z + \varepsilon^{z-1}(1 - x)^z\right) = (z - 1) \left(x^{z-1} - (\varepsilon - \varepsilon x)^{z-1}\right)$$

so the optimal value is for $x_{\text{OPT}}$ such that $(z-1)\left(x_{\text{OPT}}^{z-1} - (\varepsilon - \varepsilon x_{\text{OPT}})^{z-1}\right) = 0$, which is $x_{\text{OPT}} = \frac{\varepsilon}{\varepsilon+1}$. Hence,

$$
\begin{aligned}
\text{OPT} &= \left(\frac{\varepsilon}{\varepsilon+1}\right)^z + \varepsilon^{z-1}\left(1 - \frac{\varepsilon}{\varepsilon+1}\right)^z \\
&= \frac{\varepsilon^{z-1}}{(\varepsilon+1)^z}(\varepsilon+1) = \left(\frac{\epsilon}{1+\varepsilon}\right)^{z-1}.
\end{aligned}
$$

Since the cost of the solution having a center at $0$ is $\varepsilon^{z-1}$, is it bigger than $(1+\varepsilon)\text{OPT}$: indeed,

$$
(1+\varepsilon)\left(\frac{\epsilon}{1+\varepsilon}\right)^{z-1} < \varepsilon^{z-1}. \tag{8}
$$

Now, consider the instance $\mathcal{I}$ and the instance $\mathcal{I}'$ that has $n$ points located at $0$. let $\mathcal{A}$ be an algorithm that, with probability more than $4/5$, computes a $(1+\varepsilon)$-approximation for $(1,z)$-clustering.

Assume by contradiction that $\mathcal{A}$ samples less than $\varepsilon^{-z+1}/10$ points. Let $X$ be the random variable counting the number of points located at $1$ in that sample: we have $\Pr[X > 0] \leq \mathbb{E}[X] \leq 1/10$. So with probability at least $9/10$, $\mathcal{A}$ samples only point located at $0$: even when that event occures, $\mathcal{A}$ must output a center at a position different than $0$ (following Equation 8) with some probability $p$.

Since $\mathcal{A}$ succeeds with probability $4/5$ and $X = 0$ with probability at least $9/10$, we must have have $\frac{9}{1}0p + \frac{1}{1}0 \geq 4/5$, and so $p \geq \frac{7}{9}$.

Hence, when $\mathcal{A}$ samples only points located at $0$, it must output a center different from $0$ with probability at least $7/9$. In particular, on instance $\mathcal{I}'$, $\mathcal{A}$ fails with probability at least $7/9$, a contradiction.

So, any algorithm that computes a $(1+\varepsilon)$-approximation for $(1,z)$-clustering with probability more than $4/5$ must sample more than $\varepsilon^{-z+1}/10$ points. $\qquad\square$

## G   Improved Coreset Construction

**Lemma 13.** *Let $q$, $R_M$ and $\kappa$ be defined as above. Then for any point $c \in \mathbb{R}^d$, we have*

$$
|cost(A,c) - (cost(R_M, c) + \kappa + |A \setminus R_M| \cdot \|q - c\|^2)| \leq \varepsilon \cdot cost(A,c).
$$

*Proof.* First, we bound the difference in cost for the points that are close to $q$, i.e. the points in $R_i$ with $i \leq -1$. We have for any such point $p$

$$
\begin{aligned}
&\left|\|p-c\|^z - \|q-c\|^z\right| \\
\leq\ & \varepsilon \cdot \|p-c\|^z + \left(\frac{z+\varepsilon}{\varepsilon}\right)^{z-1}\|p-q\|^z \\
\leq\ & \varepsilon \cdot \|p-c\|^z + \left(\frac{z+\varepsilon}{\varepsilon}\right)^{z-1}\left(\frac{\varepsilon}{2z}\right)^z \cdot \frac{1}{\alpha} \cdot \Delta \\
\leq\ & \varepsilon \cdot \|p-c\|^z + \left(\frac{z+\varepsilon}{\varepsilon}\right)^{z-1}\left(\frac{\varepsilon}{2z}\right)^z \cdot \frac{1}{\alpha} \cdot \alpha \cdot \frac{cost(c)}{n} \\
\leq\ & \varepsilon \cdot \|p-c\|^z + \varepsilon \cdot \frac{cost(c)}{n}.
\end{aligned}
$$

Since there are at most $n$ in $\cup_{i \leq -1} R_i$, we therefore have

$$
\begin{aligned}
&\left|\sum_{i \leq -1}\sum_{p \in R_i} \|p-c\|^z - \|q-c\|^2\right| \\
\leq\ & \varepsilon \cdot \sum_{i \leq -1}\sum_{p \in R_i} \|p-c\|^z + \varepsilon \cdot cost(c) \\
\leq\ & 2 \cdot \varepsilon \cdot cost(c). \tag{9}
\end{aligned}
$$

Now we focus on the points in $R_i$ with $i \geq \log \varepsilon^{-z}$. We distinguish between two cases. The first case will assume that $\|q-c\|^z \leq \Delta \cdot \left(\frac{\varepsilon}{4z}\right)^{-z}$. Here, the intuition is that since these points are close to

$q$ (at least with respect to the points in $R_i$, $i \geq \log \varepsilon^{-z}$) $\kappa$ is a good approximation to their cost. The second case assumes that $\|q - c\|^z \geq \Delta \cdot \left(\frac{\varepsilon}{4z}\right)^{-z}$. Here, the intuition is that $\sum_{i \geq \log \varepsilon^{-z}} \text{cost}(R_i, c)$ is very small compared to $\text{cost}(A, c)$.

In the first case, we have for any point $p \in R_i$ with $i \geq \varepsilon^{-3z}$

$$
\begin{aligned}
\left|\|p - c\|^z - \|p - q\|^z\right| &\leq \varepsilon \cdot \|p - c\|^z + \left(\frac{z + \varepsilon}{\varepsilon}\right)^{z-1} \|c - q\|^z \\
&\leq \varepsilon \cdot \|p - c\|^z + \left(\frac{z + \varepsilon}{\varepsilon}\right)^{z-1} \cdot \Delta \cdot \left(\frac{\varepsilon}{4z}\right)^{-z} \\
&\leq \varepsilon \cdot \|p - c\|^z + \left(\frac{4z}{\varepsilon}\right)^{2z-1} \cdot \Delta \\
&\leq \varepsilon \cdot \|p - c\|^z + \varepsilon \cdot \frac{1}{\alpha} \cdot \|p - q\|^z.
\end{aligned}
$$

Again, we sum this over all points in $\cup_{i \geq \log \varepsilon^{-2z}} R_i$. We then have

$$
\begin{aligned}
&\left| \sum_{i \geq \log \varepsilon^{-2z}} \sum_{p \in R_i} (\|p - c\|^z - \|q - c\|^z) - \kappa \right| \\
={} &\left| \sum_{i \geq \log \varepsilon^{-2z}} \sum_{p \in R_i} \|p - c\|^z - \|p - q\|^z \right| + \sum_{i \geq \log \varepsilon^{-2z}} \sum_{p \in R_i} \|q - c\|^z \\
\leq{} &\varepsilon \sum_{i \geq \log \varepsilon^{-2z}} \sum_{p \in R_i} \|p - c\|^z + \sum_{i \geq \log \varepsilon^{-2z}} \sum_{p \in R_i} \varepsilon \cdot \frac{2}{\alpha} \cdot \|p - q\|^z \\
\leq{} &\varepsilon \cdot \text{cost}(A, c) + \varepsilon \cdot \frac{2}{\alpha} \cdot \text{cost}(A, q) \\
\leq{} &3\varepsilon \cdot \text{cost}(A, c) \qquad\qquad\qquad\qquad\qquad\qquad\qquad\qquad\qquad (10)
\end{aligned}
$$

We now focus on the second case. Let $A_2$ be the set of points with $\|p - q\|^z \leq 2\Delta$. Due to Markov's inequality, we have $|A_2| \geq \frac{n}{2}$. Also due to Markov's inequality, we have $\left|\bigcup_{i \geq \log \varepsilon^{-2z}} R_i\right| \leq \varepsilon^{2z} \cdot n$. We now give a lower bound on the cost of the points in $A_2$. We start by showing that the difference in cost between any point in $A_2$ and $q$ when clustering to $c$ is negligible. Since $\|q - c\| \geq \frac{4z}{\varepsilon} \cdot \Delta^{1/z}$ and $\|p - q\| \leq 2^{1/z} \cdot \Delta^{1/z}$, we have $\|p - c\|^z \geq (1 - \varepsilon)\|q - c\|^z$. This implies
$$\text{cost}(A_2, c) \geq |A_2| \cdot (1 - \varepsilon)\|q - c\|^z$$

Therefore

$$
\begin{aligned}
\sum_{i \geq \log \varepsilon^{-2z}} \text{cost}(R_i, c) &\leq \sum_{i \geq \log \varepsilon^{-2z}} (1 + \varepsilon) \cdot \text{cost}(R_i, q) + \left(\frac{\varepsilon}{4z}\right)^{2z} \cdot n \left(\frac{z + \varepsilon}{\varepsilon}\right)^{z-1} \cdot \|q - c\|^z \\
&\leq (1 + \varepsilon) \cdot \text{cost}(A, q) + \left(\frac{\varepsilon}{4z}\right)^{z+1} \cdot n \cdot \|q - c\|^z \\
&\leq (1 + \varepsilon) \cdot \text{cost}(A, q) + \left(\frac{\varepsilon}{4z}\right)^{z+1} 2|A_2| \cdot \|q - c\|^z \\
&\leq (1 + \varepsilon) \cdot 2|A_2| \cdot \Delta + \left(\frac{\varepsilon}{4z}\right)^{z+1} 2|A_2| \cdot \|q - c\|^z \\
&\leq (1 + \varepsilon) \cdot 2|A_2| \cdot \left(\frac{\varepsilon}{4z}\right)^z \cdot \|q - c\|^z \\
&\leq \varepsilon \cdot \text{cost}(A_2, c) \leq \varepsilon \cdot \text{cost}(A, c). \qquad\qquad\qquad (11)
\end{aligned}
$$

Similarly

$$
\begin{aligned}
\kappa + \sum_{i \geq \log \varepsilon^{-2z}} |R_i| \cdot \|q - c\|^z &= \sum_{i \geq \log \varepsilon^{-2z}} \sum_{p \in R_i} \|p - q\|^z + \|q - c\|^z \\
&\leq n \cdot \Delta + \varepsilon^{2z} \cdot n \cdot \|q - c\|^z \\
&\leq n \cdot \left(\frac{\varepsilon}{4z}\right)^z \cdot \|q - c\|^z \|q - c\|^z + \varepsilon^{2z} \cdot n \cdot \|q - c\|^z \\
&\leq \varepsilon \cdot \text{cost}(A_2, c) \leq \varepsilon \cdot \text{cost}(A, c). \qquad\qquad\qquad (12)
\end{aligned}
$$

Combining Equations 9, 10, 11, and 12 and rescaling $\varepsilon$ now yields the claim. $\qquad\square$

*Proof of Theorem 2.* Let $c$ be an arbitrary solution. For the points in $A \setminus R_M$, Lemma 13 deterministically allows us to bound the error by at most $\varepsilon \cdot \text{cost}(A, c)$. So we now turn our attention to the rings in $R_M$. We have for all $c \in \mathbb{R}^d$

$$
\left| \sum_{p \in R_M} \|p - c\|^z - \sum_{p \in \Omega} w_p \cdot \|p - c\|^z \right| \leq \sum_{R_i \in R_M} | \sum_{p \in R_i} \|p - c\|^z - \sum_{p \in \Omega_i} w_p \cdot \|p - c\|^z |
$$

$$
\leq \sum_{R_i \in R_M} \varepsilon \cdot (\text{cost}(R_i, q) + \text{cost}(R_i, c))
$$

$$
= \varepsilon \cdot (\text{cost}(R_M, q) + \text{cost}(R_M, c))
$$

where the second inequality uses Theorem 4. Thus, taking a union bound over all $R_i \in R_M$, we have with probability at least $1 - O(\log 1/\varepsilon)\delta$ for all points $c \in \mathbb{R}^d$

$$
\left| \sum_{p \in A} \|p - c\|^z - (\sum_{p \in S} w_p \|p - c\|^z + \kappa) \right| \leq 2\varepsilon \cdot (\text{cost}(A, c) + \text{cost}(A, q))
$$

Rescaling $\varepsilon$ by a factor $4/\alpha$ and rescaling $\delta$ by a factor $1/\log 1/\varepsilon$ yields the desired bounds. $\qquad\square$

## H  Experiments

**Implementation:**  We used a variant of Algorithm 1 which now describe. Instead of specifying a desired accuracy, the algorithm is access to $m$ samples picked uniformly at random from the data set. As an $\alpha$-approximate solution $q$, the algorithm merely selects a random point.

We also estimate $\frac{OPT}{n}$, by sampling another point $q'$ and using $\|q - q'\|^z$ as a (coarse) estimate. We then apply the pruning procedures. Our algorithms chose $\{100, 200, \ldots, 1000\}$ samples. For each sample size, we repeated the algorithm 10 times and outputted the best center we could find.

Since the objective function is convex, we use a (simple) stochastic gradient descent on both the sample and the full data set to compute a desired center. We iterated over the data set a total of 10 times. In every iteration, we partitioned the data into random chunks of size $\min(m, 2000)$, and used chunk to perform a gradient step. We did not attempt to optimize the stochastic gradient descent; as our focus is less on solving the problem in the fastest way possible and more on showcasing how the sublinear algorithm can be used to potentially speed up any baseline algorithm.

The algorithms were coded in Python and run on a Intel Core i7-8665U processor with four 2 GHz cores and 32 GByte RAM.

**Results**  Tables with exact figures are given below. Here, we report and interpret the results.

On the Road Network data set, all samples sizes found a nearly optimal solution in at least one of the 10 repetitions, with the largest deviation from the optimum of $4\%$ occurring for 500 samples and the $z = 4$ problem. In addition, the sublinear algorithms all required only a very small amount of time compared to the baseline optimal solution (e.g. a factor of at least 400 quicker for the largest sample size). What is notable is that starting with $z = 5$, the variance in cost of any given sample size increased significantly. Since this occurred regardless of sample size, we attribute this effect to quality of the seeding solution ($q$ in Algorithm 1). The approximation factor of $q$ directly impacts the quality of the subsequent coreset construction, meaning that even with large sample sizes, the algorithm has difficulty to recover. This means that good seeding solution $q$, for example using Lemma 12 is essential.

Processing the USCensus data set was in its entirety was time consuming, running for more than 90 minutes. Constructing the coreset and optimizing it never took more than 14 seconds, however the algorithm did not compute near optimal solutions as was the case for Road Networks data set. For $z = 3$ and $z = 5$ the approximation was still rather small, and tightly concentrated. For to the authors very unclear reasons, there exists a larger gap at $z = 4$. While a gap of that magnitude is consistent with the lower bound, the data set does not seem to have a structure similar to said lower bound.

| Samples | z = 3 Cost | | | Time | z = 4 Cost | | | Time | z = 5 Cost | | | Time |
|---|---|---|---|---|---|---|---|---|---|---|---|---|
| | Min | Avg | Var/Avg² | Avg | Min | Avg | Var/Avg² | Avg | Min | Avg | Var/Avg² | Avg |
| 100 | 4,90 | 6,67 | 0,08 | 0,62 | 1,79 | 2,83 | 0,12 | 0,62 | 7,47 | 10,13 | 0,35 | 0,61 |
| 200 | 4,80 | 6,36 | 0,05 | 1,16 | 1,86 | 3,85 | 0,24 | 1,29 | 7,37 | 8,29 | 0,30 | 1,16 |
| 300 | 4,79 | 7,40 | 0,21 | 1,74 | 1,78 | 2,48 | 0,16 | 1,76 | 7,39 | 12,44 | 0,15 | 1,76 |
| 400 | 4,86 | 6,89 | 0,11 | 2,27 | 1,79 | 2,82 | 0,13 | 2,37 | 7,37 | 12,44 | 0,51 | 2,41 |
| 500 | 4,95 | 7,80 | 0,04 | 2,88 | 1,84 | 2,88 | 0,47 | 2,90 | 7,39 | 11,11 | 0,61 | 2,89 |
| 600 | 4,84 | 10,35 | 0,47 | 3,42 | 1,78 | 2,91 | 0,76 | 3,46 | 7,55 | 21,52 | 0,23 | 3,48 |
| 700 | 4,79 | 6,67 | 0,22 | 3,97 | 1,80 | 3,12 | 0,31 | 4,10 | 7,37 | 16,22 | 0,60 | 4,06 |
| 800 | 4,79 | 6,92 | 0,19 | 4,60 | 1,78 | 6,20 | 0,67 | 4,62 | 7,38 | 15,89 | 0,57 | 4,71 |
| 900 | 4,79 | 9,27 | 0,58 | 5,12 | 1,78 | 3,43 | 0,69 | 5,23 | 7,37 | 25,93 | 0,20 | 5,26 |
| 1k | 4,81 | 10,97 | 0,30 | 5,65 | 1,78 | 4,91 | 0,62 | 5,90 | 7,47 | 29,68 | 0,41 | 5,81 |
| OPT | 4,79 | - | - | 871 | 1,78 | - | - | 875 | 7,37 | - | - | 877 |

| Samples | z = 6 Cost | | | Time | z = 7 Cost | | | Time | | | | |
|---|---|---|---|---|---|---|---|---|---|---|---|---|
| | Min | Avg | Var/Avg² | Avg | Min | Avg | Var/Avg² | Avg | | | | |
| 100 | 3,40 | 5,38 | 0,20 | 0,63 | 1,59 | 2,28 | 0,32 | 0,61 | | | | |
| 200 | 3,32 | 6,22 | 0,41 | 1,20 | 1,60 | 2,72 | 0,46 | 1,19 | | | | |
| 300 | 3,51 | 11,47 | 0,91 | 1,77 | 1,59 | 6,20 | 1,21 | 1,82 | | | | |
| 400 | 3,35 | 6,54 | 0,71 | 2,33 | 1,59 | 8,70 | 4,12 | 2,36 | | | | |
| 500 | 3,54 | 8,22 | 1,43 | 2,89 | 1,61 | 2,42 | 0,33 | 2,92 | | | | |
| 600 | 3,32 | 6,33 | 0,85 | 3,47 | 1,61 | 14,01 | 2,25 | 3,50 | | | | |
| 700 | 3,32 | 7,86 | 0,93 | 4,06 | 1,60 | 6,53 | 2,87 | 4,10 | | | | |
| 800 | 3,35 | 11,10 | 2,25 | 4,63 | 1,61 | 5,06 | 0,76 | 4,70 | | | | |
| 900 | 3,31 | 8,46 | 0,29 | 5,21 | 1,59 | 7,49 | 0,54 | 5,28 | | | | |
| 1k | 3,34 | 7,89 | 0,57 | 5,98 | 1,59 | 2,35 | 0,16 | 5,78 | | | | |
| OPT | 3,31 | - | - | 885 | 1,59 | - | - | 882 | | | | |

Figure 1: Overview of cost and running time for the sublinear algorithm on the Road Networks data set. Costs scaled by a factor $10^9$ for $z = 3$, $10^{11}$ for $z = 4$ and $z = 5$, $10^{14}$ for $z = 6$ and $10^{16}$ for $z = 7$. The variance was extremely small for running times, so we omit it. Running time is given in seconds. The running time for the sampling algorithms only considers the time required to sample the points, prune the data set, and run the optimization, i.e. the time required to evaluate the computed solution on the entire data set is not included.

| Samples | z = 3 Cost Min | Avg | Time Avg | z = 4 Cost Min | Avg | Time Avg | z = 5 Cost Min | Avg | Time Avg |
|---|---|---|---|---|---|---|---|---|---|
| 100 | 1,306 | 1,310 | 6,03 | 1,907 | 1,910 | 7,03 | 1,527 | 1,560 | 7,22 |
| 200 | 1,306 | 1,311 | 6,03 | 1,907 | 1,911 | 7,04 | 1,518 | 1,554 | 7,81 |
| 300 | 1,305 | 1,310 | 6,30 | 1,907 | 1,908 | 7,59 | 1,523 | 1,560 | 8,65 |
| 400 | 1,305 | 1,310 | 6,48 | 1,907 | 1,909 | 7,74 | 1,521 | 1,544 | 9,06 |
| 500 | 1,306 | 1,309 | 6,74 | 1,906 | 1,908 | 7,92 | 1,512 | 1,547 | 9,84 |
| 600 | 1,305 | 1,308 | 7,06 | 1,907 | 1,908 | 8,21 | 1,516 | 1,553 | 10,39 |
| 700 | 1,307 | 1,309 | 7,20 | 1,907 | 1,908 | 8,35 | 1,516 | 1,545 | 11,14 |
| 800 | 1,306 | 1,309 | 7,53 | 1,907 | 1,908 | 8,82 | 1,528 | 1,547 | 11,86 |
| 900 | 1,306 | 1,310 | 7,60 | 1,907 | 1,909 | 8,98 | 1,523 | 1,553 | 12,43 |
| 1k | 1,307 | 1,312 | 8,04 | 1,906 | 1,909 | 9,6 | 1,526 | 1,550 | 13,26 |
| OPT | 1,125 | - | 5544 | 1,296 | - | 5586 | 1,499 | - | 5934 |

Figure 2: Overview of cost and running time for the sublinear algorithm on the USCensus data set. Costs scaled by a factor $10^{12}$ for $z = 3$, $10^{14}$ for $z = 4$ and $10^{16}$ for $z = 5$. The variance was extremely small for all values (cost and running time), as indicated by the small gaps between minimum and average. We therefore omitted it from the table. The largest variance (relative to the squared cost) we encountered was for $z = 5$ and 600 samples, where it was still below $0.0005$. Running time is given in seconds. The running for the sampling algorithms only considers the time required to sample the points, prune the data set, and run the optimization, i.e. the time required to evaluate the computed solution on the entire data set (which vastly exceeds the given time bounds) is not included.