# OpenReview forum: "Improved Coresets and Sublinear Algorithms for Power Means in Euclidean Spaces"
_NeurIPS.cc/2021/Conference — NeurIPS 2021 Spotlight_

### Official Review · Reviewer_mXmR · 2021-07-12

**Rating:** 6
**Confidence:** 3

**Summary:**

The paper handles the following problem: Given a set of $n$ points in $R^d$, the goal is to compute in sublinear time an approximation for the point that minimizes the sum of euclidian distances to the power of $z$ over the input points.

Also, it suggests a coreset for this problem.


**Limitations And Societal Impact:**


1. Experimental results:
   a. The paper should have experimental results in its main body.

   b. Why there are no graphs at all? the reader is interested in understanding the phenomena not just looking at the numbers.

 c.  You should compare yourself to other methods, e.g.,
         --- to uniform sampling in all cases,
         --- Feldman and Langberg in the relevant cases,
         --- and many other results in the cases of z=2 (e.g., median of means).

 d. Also, when comparing your results to other methods (e.g., uniform sampling), the x-axis of the graph can be the time took to compute the approximation (including the time took to compute the small subset) not the size of the coreset.

 e. More discussion needed, e.g., when your coreset performs better in practice (at what z values), when it is better than uniform sampling and why?

2. Can you explain why it is interesting to solve the problem when z is not 2,1 or inf?

3. No conclusion or future work was provided.

**Main Review:**

The paper provider a solid theoretical contribution. It suggests a sublinear time algorithm for approximating the point that minimizes the sum of euclidian distances to the power of $z$ over a given input set of points.
The required number of points (samples) in order to obtain with probability at least $1-\delta$ a $1+\varepsilon$ approximation to the optimal solution is $O(\varepsilon^{-z-3}log^2(1/\delta)polylog(\varepsilon^{-1}))$ (generalization over prior works)

The paper also shows that we can compute a coreset for such a problem, where the size of the coreset is $O(\varepsilon^{-2}2^{O(z)})$.

Finally, they show that the exponential dependency in $z$ is necessary for computing an approximation.

To do so, they used and improve the technique introduced by Chen [7] (i.e.,  given a fine initial guess, partition the points into rings that grow exponentially according to their costs) by showing that it is sufficient to only consider O(log \varepsilon^ {−1}) rings.

They suggest an improved analysis of sampling from rings. The state-of-the-art analysis by Feldman and Langberg [15] reduces the problem of computing a coreset to sampling in a (weighted) range space of bounded VC dimension.  They improve over the previous analyses via the chaining technique.

I liked the paper, it has a solid theory and novel ideas, it's written well.
However, the experimental results section is very weak see Limitations section.

Note: I couldn't check the proofs due to time constraints and since they are long.



**Time Spent Reviewing:**

7

---

> ### Author Response · Authors · 2021-08-05
> **Reply to reviewer mXmR**
>
> 2:  The main application for $z \neq 1, 2, \infty$ is to interpolate between $z=2$ and $z= \infty$, in particular because $z= \infty$ is not feasible. Skewness (a measure of the asymmetry of the probability distribution of a real-valued random variable about its mean) and kurtosis (a measure of the "tailedness" of the probability distribution) are the centralized moments with respect to the three and the four norms and are frequently used in statistics. The power mean is a way of estimating these values for multivariate distributions. Another application is when dealing with non-Euclidean distances, such as the Hamming metric, coreset constructions  for $z$ can be reduced to coreset constructions for $2z$. So for example if we want the mean in Hamming space, we can reduce it to the $z=4$ case in squared Euclidean spaces. For a reference with more details we refer to the 2020 STOC paper "Coresets for clustering in Euclidean spaces: importance sampling is nearly optimal" by Huang and Vishnoi, as well as followup work. Another application that is not as theoretical in nature is that $z=3$ is used to determine Latex line breaks.
>
> 3: We will add a conclusion. For future work: a natural direction is to extend the chaining argument to arbitrary values of $k$, to make the overall approach work for $k$-means, $k$-median, etc.. Also, there is still a gap in the exponent of $\varepsilon$ between the lower and upper bounds in the sublinear algorithm. Closing this gap is a nice open problem.
>
> 1a: This is right, we will add the experimental results to the main body, thanks!
>
> 1b: We understand that graphs could help the reader understand better our results and will add them to the camera ready, thanks!
>
> 1d: We had decided to use the number of samples as x-axis because this is supposed to be the main dependency for the running time. We will complement this with a plot of the actual running time instead.
>
> 1c, 1e: We did not conduct extensive experiments for two reasons: (1) All sublinear algorithms are just uniform sampling and it is known that adaptivity does not help in the context of sublinear algorithms. So there is not much variation in the known approaches to make meaningful experiments. This work can be essentially seen as defining a baseline. (2) The importance sampling distributions used by coresets in general cannot be used by sublinear algorithms. For coresets, the state of the art coreset sampling distributions (in particular also Feldman and Langberg) are extremely similar, despite the analysis having little resemblance. Moreover, evaluating the guarantee of strong coresets is a hard problem. One coreset could preserve the cost of one particular solution well, while preserving the cost of another very poorly. It is an interesting and hard open problem to find a way to efficiently evaluate the quality of a coreset.
>
> We will incorporate the above comments to the paper. We thank you very much for the time spent on the paper and the comments that would indeed improve the quality of the paper.

---

> > ### Comment · Reviewer_mXmR · 2021-08-09
> > **Thanks for the reply**
> >
> > Thank you for clarifying question 2, and adapting my suggestions.
> > Please add this explanation to the introduction as it is part of your main contribution.

---

### Official Review · Reviewer_av2R · 2021-07-16

**Rating:** 7
**Confidence:** 4

**Summary:**

The paper studies sublinear algorithms for Power means i.e., given a set of $n$ points $a_1,\ldots, a_n$, compute a center $c$ that minimizes (upto $1+\varepsilon$) factor the objective $\sum_{i = 1}^n d(a_i, c)^z$ for any $z \ge 1$. The authors show that $\varepsilon^{-z-3}$ uniformly random samples is enough to achieve a $1+\varepsilon$ approximation. They also construct coresets of size $\varepsilon^{-2}$.

**Main Review:**

They proceed to prove that the uniform sampling works as follows:
- $q$ be an approximate solution
- Divide the $n$ points into rings based on their distance from $q$
- They argue that if the rings are very sparse, then they can be ignored as they won't contribute to the cost much
- For rings that are dense enough, uniform sampling for that ring approximates the cost well.
- To prove that uniform sampling from a ring works to approximate cost well, they consider a net over the vector of distances from a point $c$ to the set of points in $R$. They only require that the net-vectors approximate the distances of points $p$ in the ring such that $d(p, c)$ is comparable to $d(p, q)$. They then obtain a bound on the size of the net and then argue using a chaining argument that uniform sampling from the ring approximates cost of the ring with respect to any potential center $c$.

Strengths:
- Important problem for which algorithms weren't known for general $z$
- Obtain sizes that are close to optimal using a simple and nice lowerbound

Weaknesses:
- Paper is very poorly written which makes it harder to understand what is actually going on. For example, I don't understand what is happening in the expectation in line 191. The authors take expectation of supremum over all the centres $c$, but then use $v$ in the expectation and then say for every $v$ in the net. Do you have to prove the bound for every $v$ in the net or only for $v$  in the net that is close to distance vector of center $c$ or is it a typo?
- There definitely is novelty in their algorithms and proof techniques but the paper in its current form is very hard to be reviewed and check for bugs which makes me recommend not accepting the paper.

Suggestions:
- I believe once the paper is written well and easy to comprehend, the results are definitely strong enough to be accepted at a top place. I'd suggest you rewrite the Techniques section to give a much better overview of the algorithms and better explain how various lemmas help in proving the theorems.

**Time Spent Reviewing:**

5

---

> ### Author Response · Authors · 2021-08-05
> **Reply to reviewer av2R**
>
> Thank you very much for the time spent on the paper and the comments.
> We appreciate the difficulty of reading the paper. Unfortunately, we do not believe that we can come up with a significantly simpler "zero calorie" proof, and the technical novelty of our paper will make it hard to obtain an "easy-to-understand" proof. We will improve the quality of the exposition in the camera ready version to make it as easy as possible.
> To address the specific comment of line 191: the interpretation that the net points v are close to c is what we intended. The supremum taken over c instead of v was a typo which we can correct and we apologize for that. The correct version is also what is more formally stated in lemma 4, in line 198-199 below.
>
> We have also made passes over the paper to find and eliminate similar typos.

---

> > ### Comment · Reviewer_av2R · 2021-08-05
> > **Few Clarifications**
> >
> > In the paragraph beginning at line 102, you say that the idea is to use a sequence of solutions $c_h$ that better approximate a candidate solution. Where in the theorems/lemmas in the paper do you use this? As I understand, the proof is as follows:
> > 1. With good probability, the $q$ you sample is a good solution
> > 2. Divide the points into rings centered at point $q$
> > 3. The argument is that either rings can be ignored or snapped to $q$ or that uniform sampling works
> > 4. To show that uniform sampling works for a ring with large enough number of points, you use a net argument.
> >
> > Am I missing anything? Can you briefly explain what the paragraphs beginning at line 102 are trying to convey?
> >
> > Also, in line 245, you say that $q$ is an $\alpha$ approximation with probability $1/\alpha$. You may have meant with probability $\ge 1 - 1/\alpha$. Even then, the statement is clear to me. If the objective is as defined, then I was able to prove that with probability $\ge 1 - \alpha$, $q$ is a $2^z(1+\alpha)$ approximation and not that it is an $\alpha$ approximation. Am I missing something here?
> >
> > I am not looking for a "zero-calorie" proof per se. I just wished that the Techniques section could use a bit more work. It might help if you point to the specific lemmas or theorems when talking about them in the Techniques section. I understand that it may be hard to properly explain your Techniques given the page constraints.

---

> > > ### Author Response · Authors · 2021-08-05
> > > **Addressing the comments.**
> > >
> > > Thanks a lot for the quick reply, giving us the opportunity to further
> > > clarify.
> > > We would like to stress that we truly have two results. One is the
> > > sublinear algorithm. The other is an improved coreset construction for
> > > specific types of input points. We show how to leverage the coreset
> > > construction for the constrained case to both general inputs, as well as
> > > to the sublinear algorithm. The exposition following line 102 describes how to
> > > achieve the coreset result for the constrained inputs.
> > >
> > > Previously, coresets for powers of distances (other than for the mean,
> > > for which an $\varepsilon^{-2}$ sized coreset is implicit in Feldman,
> > > Schmidt, and Sohler) in Euclidean spaces had space requirements of
> > > either $d\cdot \varepsilon^{-2}$ (see for example the seminal
> > > Feldman/Langberg paper or later results by Bachem, Lattanzi, and Lucic)
> > > or $\varepsilon^{-4}$ (e.g. recent work by Braverman, Jiang,
> > > Krauthgamer, and Wu or Cohen-Addad, Saulpic, and Schwiegelshohn). We
> > > improve this to $\varepsilon^{-2}$ and to the best of our knowledge we
> > > are the first to break the $\varepsilon{-2}\cdot
> > > \min(d,\varepsilon^{-2})$ barrier for powers other than the mean.
> > > The bottleneck where the sizes of the nets when applying a union bound.
> > > The paragraphs following line 102 describe how we do this. For the
> > > application to sublinear algorithms, this is not necessary if one were
> > > satisfied with $\varepsilon^{-z-5}$ samples, and using a different
> > > analysis, or even defaulting to a previous analysis.
> > > But for coresets, we believe that this improvement is substantial.
> > >
> > > As for the second comment: Indeed, it should be a $2^{z} \cdot \alpha$
> > > approximation with probability $1-1/\alpha$, thank you for pointing this
> > > out. This was a very unfortunate typo. It does not affect the bounds of
> > > the paper in any way. The analysis of finding a good seeding solution is
> > > done in more detail in lemma 12.

---

> > > > ### Comment · Reviewer_av2R · 2021-08-05
> > > > **Thanks for Clarification.**
> > > >
> > > > So, the union bound is over all the rings as mentioned in line 669, right? I am just confused about the comment that you consider sequence of solutions $c_h$ that approximate candidate solution as $h \rightarrow \infty$.
> > > >
> > > > I will update my scores for the paper. Please correct the mistakes in the final version of the paper and try to make "Techniques" part of the paper better present your ideas.

---

> > > > > ### Author Response · Authors · 2021-08-05
> > > > > **Addressing Comment**
> > > > >
> > > > > Again, thank you so much for the quick reply.
> > > > >
> > > > > There are multiple instances of the union bound. Line 669 addresses the union bound succeeding for every ring consisting of an easier instance in the sense that points are more or less equidistant to the initial solution.
> > > > >
> > > > > The other union bound, which we attempt to improve on, involves enumerating all candidate centers. We hope the following comments will make it more clear. If you consider these comments helpful, we will modify the Techniques section along these lines.
> > > > >
> > > > > The standard way to prove a coreset guarantee is to show that using X samples we preserve the cost of a single solution with probability 1-delta. If there exist T solutions then we set delta = 1/T (increasing the size of X) and have obtained a coreset, which we call the "naive" union bound. This works in certain cases such as finite metrics, but is insufficient if we have infinitely many solutions such as Euclidean spaces.
> > > > > The simplest way to improve over the naive union bound is to discretize the space and then apply the naive union bound on the discritization. In literature this is sometimes called an $\varepsilon$-net bound. Again, this can be optimal or close to optimal for certain metrics, but so far these arguments have only lead to the $\varepsilon^{-2}\cdot \min(d, \varepsilon^{-2})$ bounds for coresets in Euclidean spaces. This is the union bound we aim to beat.
> > > > >
> > > > > For completeness, we also added comments on how chaining allows to improve on the two aforementioned approaches. Similar comments are included in line 102 and the following and we will do our best to make this exposition clearer. Hopefully in the context of this comment, it will explain the ideas a bit better.
> > > > >
> > > > > Instead of applying a union bound over the discretization in "one shot", we apply a union bounded over a nested sequence of increasingly better discretizations. Essentially, instead of only using a center $c^{\log \varepsilon^{-1}}$ as a substitute for $c$, we use centers $c^h$ for different values of $h\in \{0,\ldots \log 1/\varepsilon}$. We can then write, for any input point set P, $\sum_{p\in P}\text{cost}(p,c) = \sum_{h\geq 0} \sum_{p\in P}\text{cost}(p,c^{h+1})-\text{cost}(p,c^h)$, where $\text{cost}(p,c^0)$ is defined to be = 0. We now only apply the naive union bound for successive summands, i.e. we approximate $\sum_{p\in P}\text{cost}(p,c^{h+1})-\text{cost}(p,c^h)$ up to an error of $\varepsilon\cdot \sum_{p\in P}\text{cost}(p,c)$.
> > > > >
> > > > > To make this framework work, one has to take special care in constructing the nets, for example to ensure that the sum converges rapidly so that we do not lose $\varepsilon\cdot \sum_{p\in P}\text{cost}(p,c)$ too many times, and also the type of nets require somewhat stronger than a standard $\varepsilon$-net of a set of unit vectors.

---

> > > > > > ### Comment · Reviewer_av2R · 2021-08-05
> > > > > > **Thanks**
> > > > > >
> > > > > > Thanks for the clarification! I saw the proof Lemma 6 where you use this union bound.

---

### Official Review · Reviewer_FUAC · 2021-07-21

**Rating:** 8
**Confidence:** 3

**Summary:**

The paper gives sub-linear time algorithms for approximating ($1+\epsilon$ approximation) the centroid of a set of points in $d$-dimensional real space, under a distance function which is a power $z$ of the Euclidean distance. So, in particular, for $z=1$ the problem is estimating the Fermat-Weber point, and for $z=2$ the problem is estimating the mean. The algorithms use a random sample of the point set. Moreover, the paper also gives a probabilistic construction of a coreset of size independent of the dimension $d$ that estimates well the cost of any candidate center. The algorithms employ an interesting modification of an algorithm by Chen, as well as dimension reduction and a probabilistic analysis using chaining.

**Main Review:**

The sub-linear time algorithms for $z=1$ and for $z=2$ in fact do not improve upon previous work (the authors note that). In fact, for $z=2$ a rather simple probabilistic analysis would show that sampling (with replacement) uniformly at random $O(1/\epsilon^2)$ points yields a good estimate of the mean. Thus, the main contribution here is for $z > 2$. It would be useful to point out applications of larger $z$. That there exists a coreset of size $\tilde{O}(\exp(z)/\epsilon^2 )$ I believe was not known (but independence of the dimension was known). So may take is that this is the main contribution of the paper, improving the coreset size from $\tilde{O}(\exp(z)/\epsilon^4)$ to $\tilde{O}(\exp(z)/\epsilon^2)$. This contribution is substantial and thus the paper is interesting.

**Time Spent Reviewing:**

4 hours

---

> ### Author Response · Authors · 2021-08-05
> **Reply to reviewer FUAC**
>
> Thank you very much for the time spent on the paper and the comments. We will add more details on the  applications for $z \neq 1, 2, \infty$ to the paper, as detailed in the comment we made to reviewer mXmR.

---

> > ### Comment · Reviewer_FUAC · 2021-08-08
> > **examples**
> >
> > I read your comment to the other reviewer. Indeed, the list of applications is adequate. I think that they should be included in the discussion in the introduction, because your primary contribution is for $z\ne 1, 2$. Also, the mention of $z = \infty$ is somewhat misleading because the exponential dependence on $z$ prevents a solution for $z = \infty$ (that would have been a very surprising result). However, as motivation for fixed $z > 2$ (as an approximation) this is more convincing.

---

### Official Review · Reviewer_isuV · 2021-08-08

**Rating:** 6
**Confidence:** 5

**Summary:**

Given n points in R^d, the goal is to compute a center x in R^d that minimizes the sum (dist(p,x))^z of distance to the power of z over every input point p. The main (sub-linear time) algorithm uniformly sample ~1/eps^z points and use smart processing to obtain a (1+epsilon) approximation to the desired center x. Nice application is differential privacy. Lower bounds are also provided. More generally, the authors suggest such coreset that approximates every query center.




**Ethical Concerns:**

OK

**Ethics Review Area:**

["I don’t know"]

**Limitations And Societal Impact:**

OK

**Main Review:**

I already reviewed a previous version of this paper. Corrections and references were added.
There are still some concerns.

Strengths:
- Fundamental problem in machine learning and optimization
- Clear writing
- Open Python code which is not so common these days and will be appreciated by the non-theory guys
- Interesting non-trivial algorithm
- Novel approach with potential many applications.

Weaknesses:
I am mainly concern with relation to existing work and novelty.
For example, it is claimed that their coreset of size ~O(1/eps^2) improves the coreset of ~O(d/eps^2) by Feldman and Langberg  [STOC'11]. However, [STOC'11] also suggested how to remove the d using the fact that the optimal center can be approximated by ~1/epsilon points, which implies a VC-dimension that is independent of d.
Also, when the authors write O(1/eps^2) instead of 2^{100z}*O(1/eps^2) that should mention that z is fixed.



**Time Spent Reviewing:**

4 (including last times)

---

> ### Author Response · Authors · 2021-08-08
> **Addressing Comment**
>
> We thank the reviewer for the reply.
>
> Regarding the novelty, we believe that reviewer may be underestimating the contribution. The fact that an optimal center can be approximated with $1/\varepsilon$ points does not imply that the VC dimension is independent of $d$. Indeed, many previous coreset constructions (listed below) bypass the $d/\varepsilon^{2}$ by Feldman and Langberg bound only with difficulty. The fact that a small set of points is enough to preserve the cost only results in "weak" or "streaming" coresets, to use the notation by Feldman and Langberg. These coresets only preserve the optimum as opposed to approximating the cost for all solutions and, at least as far as we can tell, have a dependency of at least $\varepsilon^{-3}$. Moreover, the improvements only trade off the factor $d$ with a factor $\varepsilon^{-2}$ in the best case. Hence, these previous bounds were natural barriers.
>
> As for the remark on the exponential dependency in $z$: We will correct the mentions of them. We did not intend to hide this dependency at all and apologize if it seemed like we mislead the reader.
>
> Finally, there is no application to differential privacy.
>
> Bounds improving the dependency on $d$:
> 1. Feldman, Schmidt, Sohler (SODA 2013, Sicomp 2020): Introducing the coreset with offset definition, they observe that the mean is enough to obtain a coreset. It does not apply to other powers.
> 2. Sohler, Woodruff (FOCS 2018): First dimension free coresets for powers. The exponent of $\varepsilon^{-1}$ is at least $4$, and the running time is exponential in $\varepsilon^{-1}$, even for a single center.
> 3. Huang, Vishnoi (STOC 2020): Application of terminal embeddings, resulting in coresets for powers independent of the dimension in polynomial time. See also follow-up work by Braverman, Jiang, Krauthgamer, and Wu (SODA 2021) which results in better bounds in some regimes.
> 4. Feng, Kacham, Woodruff (ICML 2021): Constructive version of the Sohler-Woodruff algorithm.
>
> None of these works, except for Feldman, Schmidt and Sohler for the mean, have managed to improve the bound for strong coresets beyond $\varepsilon^{-4}$, even for a single center.

---

> > ### Comment · Reviewer_isuV · 2021-08-08
> > **Weak coresets**
> >
> > "The fact that a small set of points is enough to preserve the cost only results in "weak" or "streaming" coresets"
> >
> > Indeed, your first and main result (in the abstract and Theorem 1) only extracts an approximated solution and not a strong coreset. Can you please answer my comment with respect to this result?

---

> > > ### Comment · Reviewer_isuV · 2021-08-08
> > > **Strong coresets**
> > >
> > > Most of the citations you mentioned aim to solve the k-median problem, which is much harder. In particular, uniform sample clearly won't work.
> > >
> > > "Sohler, Woodruff (FOCS 2018): First dimension free coresets for powers. The exponent of  is at least 4, and the running time is exponential in , even for a single center."
> > >
> > > The existence of a strong coreset of size independent of d implies VC-dimension that is independent of d for every query center as in your Theorem 2. Doesn't it? Then we can trivially use eps-sample as in Feldman and Langberg.
> > >
> > > BTW, the running time was improved to~O(nnz) by Feng et al.:
> > > https://arxiv.org/abs/1912.12003

---

> > > > ### Author Response · Authors · 2021-08-08
> > > > **Addressing Comment**
> > > >
> > > > We thank you for the quick reply.
> > > >
> > > > Regarding your first comment:
> > > > Our second theorem (theorem 2 and the last paragraph of the abstract) states that we obtain a strong coreset. We only use a weaker version to obtain optimal query complexity for the sublinear algorithm. In the normal, non-sublinear case, we compute a strong coreset with offset. If a non-zero offset is not considered acceptable, we can easily make the offset zero as well. It would be interesting to show that obtaining a coreset in the sublinear setting is also possible with $\varepsilon^{-z-O(1)}$ queries. Using our work, one would only be able to show such a result with $\varepsilon^{-2z-O(1)}$ samples, which may or may not be optimal.
> > > >
> > > > Regarding your second comment:
> > > > It is not clear that a coreset of size $O(\varepsilon^{-2})$ implies that the VC dimension is bounded. But even if there were such a connection, VC-dimension is not equivalent to coreset-size. Indeed, bounded VC dimension does not imply bounded doubling dimension, or vice-versa. However, there exist coresets in bounded doubling dimension (see for example Huang, Jiang, Li, Wu, FOCS 2018). Another example is $k$-center, for which the VC dimension is similarly $k\cdot d\log k$, but coresets even for $k=1$ require $\Omega(\varepsilon^{-d/2})$ many points.
> > > >
> > > > We are aware of the fact that $k$-median is harder in general. However, we do not use uniform sampling to construct the coreset. Uniform sampling is only used for the sublinear algorithm. When used to construct a coreset, we use a variant of Chen's sampling procedure. Moreover, Chen's sampling itself is a precursor to sensitivity sampling. In Chen's version, this resulted in weaker bounds, such as a dependency on $\log n$. We manage to avoid these.
> > > >
> > > > We believe that the Feldman and Langberg algorithm could be used to obtain coresets of size $\tilde{O}(\varepsilon^{-2})$. However, this does not follow from VC dimension. We believe it would be preferable to show that "normal" sensitivity sampling can be used (aside from the application to the sublinear algorithm), instead of our algorithm, however, this requires a proof and it does not trivially follow from our work.
> > > > Indeed, if the range space (or function) space is considered to be the intersection of $k$ halfspaces in $d$ dimensions, Csikos, Mustafa, Kupavskii (J. Mach. Learn. Res. 2019) showed a lower bound of $\Omega(kd\log k)$ on the VC dimension. While one could consider a different range space, this is the one that Feldman and Langberg and therefore all clustering papers building on the Feldman and Langberg framework are considering.
> > > >
> > > > Finally, we mentioned Feng et al in our previous comment in item 4.

---

### Decision · Program_Chairs · 2021-09-27

**Decision:**

Accept (Spotlight)

**Comment:**

This is a strong paper making contributions to sublinear time algorithms and coresets for power means clustering. All reviewers agree it should be accepted, and I think it is a candidate spotlight paper.